# REGULARIZED DISTRIBUTION MATCHING DISTILLATION FOR ONE-STEP UNPAIRED IMAGE-TO-IMAGE TRANSLATION

## ABSTRACT

Diffusion-based generative models achieve SOTA results in mode coverage and generation quality but suffer from inefficient sampling. Recently introduced diffusion distillation techniques approach this issue by transforming the original multi-step model into a one-step generator with approximately the same output distribution. Among these methods, Distribution Matching Distillation (DMD) offers a suitable framework for training general-form one-step generators, applicable beyond unconditional generation. In this paper, we propose a modification of DMD, called Regularized Distribution Matching Distillation (RDMD), which applies to the unpaired image-to-image (I2I) translation problem. To achieve this, we regularize the generator objective from DMD by adding the transport cost between its input and output. We validate the method's applicability in theory by establishing its connection with optimal transport. Moreover, we demonstrate its empirical performance in application to several translation tasks, including 2D examples and I2I between different image datasets, where it performs on par or better than multi-step diffusion baselines.

## 1 INTRODUCTION

One of the global problems of contemporary generative modeling consists of solving the so-called generative learning trilemma (Xiao et al., 2021). It states that a perfect generative model should possess three desirable properties: high generation quality, mode coverage/diversity of samples and efficient inference. Today, most model families tend to have only 2 of the 3. Generative Adversarial Networks (GANs) (Goodfellow et al., 2014) have fast inference and produce high-quality samples but tend to underrepresent some modes of the data set (Metz et al., 2016; Arjovsky et al., 2017). Variational Autoencoders (VAEs) (Kingma & Welling, 2013; Rezende et al., 2014) efficiently produce diverse samples while suffering from insufficient generation quality. Finally, diffusion-based generative models (Ho et al., 2020; Song et al., 2020; Dhariwal & Nichol, 2021; Karras et al., 2022) achieve SOTA generative metrics and visual quality yet require running a high-cost multi-step inference procedure.

Satisfying these three properties is essential in numerous generative computer vision tasks beyond unconditional generation. One is image-to-image (I2I) translation (Isola et al., 2017; Zhu et al., 2017), which consists of learning a mapping between two distributions that preserves the cross-domain properties of an input object while appropriately changing its source-domain features to match the target. Most examples, like transforming cats into dogs (Choi et al., 2020) or human faces into anime (Korotin et al., 2022) belong to the *unpaired* I2I because they do not assume ground truth pairs of objects in the data set. As in unconditional generation, unpaired I2I methods were previously centered around GANs (Huang et al., 2018; Park et al., 2020; Choi et al., 2020; Zheng et al., 2022), but now tend to be competed and surpassed by diffusion-based counterparts (Choi et al., 2021; Meng et al., 2021; Zhao et al., 2022; Wu & De la Torre, 2023). Most of these methods build on top of the original diffusion sampling procedure and tend to have high generation time as a consequence.

Since diffusion models succeed in both desirable qualitative properties of the trilemma, one could theoretically obtain samples of the desired quality level given sufficient computational resources. It

makes the acceleration of diffusion models an appealing approach to satisfy all of the aforementioned requirements, including efficient inference.

Recently introduced diffusion distillation techniques (Song et al., 2023; Kim et al., 2023b; Sauer et al., 2023) address this challenge by compressing diffusion models into one-step students with (hopefully) similar qualitative and quantitative properties. Among them, Distribution Matching Distillation (DMD) (Yin et al., 2023; Nguyen & Tran, 2023) offers an expressive and general framework for training free-form generators based on techniques initially introduced for text-to-3D (Poole et al., 2022; Wang et al., 2024). *Free-form* here means that the method does not make any assumptions about the generator's structure and distribution at the input. This crucial observation opens a large space for its applications beyond the $noise \rightarrow data$ problems.

In this work, we introduce the modification of DMD, called *Regularized Distribution Matching Distillation* (RDMD), that applies to the unpaired I2I problems. To achieve this, we replace the generator's input noise with the source data samples to further translate them into the target. We maintain correspondence between the generator's input and output by regularizing the objective with the transport cost between them. As our main contributions, we

1. Propose a one-step diffusion-based method for unpaired I2I;

2. Theoretically verify it by establishing its connection with optimal transport (Villani et al., 2009; Peyré et al., 2019);

3. Ablate its qualitative properties and demonstrate its generation quality on 2D and image-to-image examples, where it obtains comparable or better results than the multi-step counterparts.

## 2 BACKGROUND

### 2.1 DIFFUSION MODELS

Diffusion models (Song & Ermon, 2019; Ho et al., 2020) are a class of models that sequentially perturb data distribution $p^{\text{data}}$ with noise, transforming it into a tractable unstructured distribution, which contains no information about the initial domain. Using this distribution as a prior and reversing the process by progressively removing the noise yields a sampling procedure from $p^{\text{data}}$.

A common way to formalize diffusion models consists in defining distribution dynamics $\{p_t(\boldsymbol{x}_t)\}_{t \in [0,T]}$, obtained by adding an independent Gaussian noise $\sigma_t \varepsilon$ with progressively growing variance $\sigma_t^2$ to the original data sample $\boldsymbol{x}_0 \sim p^{\text{data}}$: $\boldsymbol{x}_t = \boldsymbol{x}_0 + \sigma_t \varepsilon$.

Conveniently, the equivalent distribution dynamics can be represented via a deterministic counterpart given by the ordinary differential equation (ODE), which yields the same marginal distributions $p_t(\boldsymbol{x}_t)$, given the same initial distribution $p_0(\boldsymbol{x}_0) = p^{\text{data}}(\boldsymbol{x}_0)$:

$$\mathrm{d}\boldsymbol{x}_t = -\frac{1}{2} \left( \sigma_t^2 \right)' \cdot \nabla_{\boldsymbol{x}_t} \log p_t(\boldsymbol{x}_t) \mathrm{d}t, \tag{1}$$

where $\nabla_{\boldsymbol{x}_t} \log p_t(\boldsymbol{x}_t)$ is called the *score function* of $p_t(\boldsymbol{x}_t)$. Equation 1 is also called Probability Flow ODE (PF-ODE). The ODE formulation allows us to obtain a *backward* process of data generation by simply reversing the velocity of the particle. In particular, one can obtain samples from $p^{\text{data}}$ by taking $\boldsymbol{x}_T \sim p_T$ and running the PF-ODE backwards in time, given access to the score function. The sampling procedure is essentially multi-step, which imposes computational challenges but allows to control the resources-quality trade-off.

Diffusion models learn score functions $\nabla_{\boldsymbol{x}_t} \log p_t(\boldsymbol{x}_t)$ of noisy distributions by approximating them via the Denoising Score Matching (Vincent, 2011) objective:

$$\int_0^T \beta_t \, \mathbb{E}_{p_{0,t}(\boldsymbol{x}_0, \boldsymbol{x}_t)} \| \boldsymbol{s}_t^\theta(\boldsymbol{x}_t) - \nabla_{\boldsymbol{x}_t} \log p_{t|0}(\boldsymbol{x}_t | \boldsymbol{x}_0) \|^2 \mathrm{d}t \rightarrow \min_\theta, \tag{2}$$

where $\beta_t$ is some positive weighting function. The minimum in the Eq. 2 is obtained at $\boldsymbol{s}_t^\theta(\boldsymbol{x}_t) = \nabla_{\boldsymbol{x}_t} \log p_t(\boldsymbol{x}_t)$. In case of the dynamics defined in the Eq. 1, conditional distributions $p_{t|0}(\boldsymbol{x}_t | \boldsymbol{x}_0)$ are equal to $\mathcal{N}(\boldsymbol{x}_t | \boldsymbol{x}_0, \sigma_t^2 I)$, which yields tractable conditional score functions $\nabla_{\boldsymbol{x}_t} \log p_{t|0}(\boldsymbol{x}_t | \boldsymbol{x}_0)$.

Given a suitable parameterization of the score network, the DSM objective is equivalent to

$$\int_0^T \beta_t \, \mathbb{E}_{p_{0,t}(\boldsymbol{x}_0, \boldsymbol{x}_t)} \| D_t^\theta(\boldsymbol{x}_t) - \boldsymbol{x}_0 \|^2 \mathrm{d}t \to \min_\theta, \tag{3}$$

where $D_t^\theta$ is called the denoising network (or simply denoiser) and is related to the score network via $\boldsymbol{s}_t^\theta(\boldsymbol{x}_t) = \left( \boldsymbol{x}_t - D_t^\theta(\boldsymbol{x}_t) \right) / \sigma_t^2$. Therefore, training diffusion models involves learning to denoise images at various noise levels.

## 2.2 Distribution Matching Distillation

Distribution Matching Distillation (Yin et al., 2023) is the core technique of this paper. Essentially, it aims to train a generator $G_\theta(\boldsymbol{z})$ to match the given distribution $p^{\text{real}}$. Its input $\boldsymbol{z}$ is assumed to come from a tractable input distribution $p^{\text{noise}}$. Formally, matching two distributions can be achieved by optimizing the KL divergence between the distribution $p^{G_\theta}$ of $G_\theta(\boldsymbol{z})$ and the data distribution $p^{\text{real}}$:

$$\mathrm{KL}(p^{G_\theta} \| p^{\text{real}}) = \mathbb{E}_{p^{\text{noise}}(\boldsymbol{z})} \log \frac{p^{G_\theta}(G_\theta(\boldsymbol{z}))}{p^{\text{real}}(G_\theta(\boldsymbol{z}))} \to \min_\theta. \tag{4}$$

Differentiating it by the parameters $\theta$, using the chain rule, one encounters a summand, containing the difference $\boldsymbol{s}^{G_\theta}(G_\theta(\boldsymbol{z})) - \boldsymbol{s}^{\text{real}}(G_\theta(\boldsymbol{z}))$ between the score functions of the corresponding distributions [1]. The pure data score function can be very non-smooth due to the Manifold Hypothesis (Tenenbaum et al., 2000) and is generally difficult to train (Song & Ermon, 2019); therefore, the authors address this challenge using the diffusion framework. To this end, they relax the original loss by using an ensemble of KL divergences between distributions, which are perturbed by the forward diffusion process:

$$\int_0^T \omega_t \, \mathrm{KL}\Big( p_t^{G_\theta} \| p_t^{\text{real}} \Big) \mathrm{d}t = \int_0^T \omega_t \, \mathbb{E}_{\mathcal{N}(\varepsilon|0,I)p^{\text{noise}}(\boldsymbol{z})} \log \frac{p_t^{G_\theta}(G_\theta(\boldsymbol{z}) + \sigma_t \varepsilon)}{p_t^{\text{real}}(G_\theta(\boldsymbol{z}) + \sigma_t \varepsilon)} \, \mathrm{d}t. \tag{5}$$

Here, $\omega_t$ is a weighting function, $p_t^{G_\theta}$ and $p_t^{\text{real}}$ are the perturbed versions of the generator distribution and $p^{\text{real}}$ up to the time step $t$. In theory, the minima of Eq. 5 objective coincides(Wang et al., 2024, Thm. 1) with the original minima from Eq. 4. Meanwhile, in practice, taking the gradient of the new loss results in the difference $\boldsymbol{s}_t^{G_\theta}(G_\theta(\boldsymbol{z}) + \sigma_t \varepsilon) - \boldsymbol{s}_t^{\text{real}}(G_\theta(\boldsymbol{z}) + \sigma_t \varepsilon)$, which can be approximated using diffusion models.

Given this, authors approximate $\boldsymbol{s}_t^{\text{real}}$ with the pre-trained diffusion model, which we will denote $\boldsymbol{s}_t^{\text{real}}$ as well with a slight abuse of notation. The whole procedure now can be considered as distillation of $\boldsymbol{s}_t^{\text{real}}$ into $G_\theta$. At the same time, $\boldsymbol{s}_t^{G_\theta}$ represents the score of the noised distribution of the generator, which is intractable and is therefore approximated by an additional «fake» diffusion model $\boldsymbol{s}_t^\phi$ and the corresponding denoiser $D_t^\phi$. It is trained on the standard denoising score matching objective with the generator's samples at the input. The joint training procedure is essentially the coordinate descent

$$\begin{cases} \int\limits_0^T \omega_t \, \mathbb{E}_{\mathcal{N}(\varepsilon|0,I)p^{\text{noise}}(\boldsymbol{z})} \log \frac{p_t^\phi(G_\theta(\boldsymbol{z}) + \sigma_t \varepsilon)}{p_t^{\text{real}}(G_\theta(\boldsymbol{z}) + \sigma_t \varepsilon)} \, \mathrm{d}t \to \min_\theta; \\ \int\limits_0^T \beta_t \, \mathbb{E}_{\mathcal{N}(\varepsilon|0,I)p^{\text{noise}}(\boldsymbol{z})} \| D_t^\phi(G_\theta(\boldsymbol{z}) + \sigma_t \varepsilon) - G_\theta(\boldsymbol{z}) \|^2 \, \mathrm{d}t \to \min_\phi, \end{cases} \tag{6}$$

where the stochastic gradient with respect to the fake network parameters $\phi$ is calculated by backpropagation and the generator's stochastic gradient is calculated directly as

$$\omega_t \left( \boldsymbol{s}_t^\phi (G_\theta(\boldsymbol{z}) + \sigma_t \varepsilon) - \boldsymbol{s}_t^{\text{real}} (G_\theta(\boldsymbol{z}) + \sigma_t \varepsilon) \right) \nabla_\theta G_\theta(\boldsymbol{z}). \tag{7}$$

Minimization of the fake network's objective ensures $\boldsymbol{s}_t^\phi = \boldsymbol{s}_t^{G_\theta} \Leftrightarrow p_t^\phi = p_t^{G_\theta}$. Under this condition, the generator's objective is equal to the original ensemble of KL divergences from Eq. 5, minimizing which solves the initial problem and implies $p^{G_\theta} = p^{\text{real}}$.

---

[1]Note that there is one more summand, which contains the parametric score $\nabla_\theta \log p^{G_\theta}$. However, its expected value is zero (Williams, 1992), and the summand can be omitted.

## 2.3 UNPAIRED I2I AND OPTIMAL TRANSPORT

The problem of unpaired I2I consists of learning a mapping $G$ between the *source* distribution $p^{\mathcal{S}}$ and the *target* distribution $p^{\mathcal{T}}$ given the corresponding independent data sets of samples. When optimized, the mapping should appropriately adapt $G(\boldsymbol{x})$ to the target distribution $p^{\mathcal{T}}$, while preserving the input's cross-domain features. However, at first glance, it is unclear what the preservation of cross-domain properties should look like.

One way to formalize this is by introducing the notion of a "transportation cost" $c(\cdot, \cdot)$ between the generator's input and output and stating that it should not be too large on average. In a practical I2I setting, we can choose $c(\cdot, \cdot)$ as any reasonable distance between images or their features that we aim to preserve, such as pixel-wise distance or the difference between embeddings.

Monge optimal transport (OT) problem (Villani et al., 2009; Santambrogio, 2015) follows this reasoning and aims at finding the mapping with the least average transport cost among all the mappings that fit the target $p^{\mathcal{T}}$:

$$\inf_{G} \left\{ \mathbb{E}_{p^{\mathcal{S}}(\boldsymbol{x})} c(\boldsymbol{x}, G(\boldsymbol{x})) \mid G(\boldsymbol{x}) \sim p^{\mathcal{T}} \right\}, \tag{8}$$

which can be seen as a mathematical formalization of the I2I task.

Under mild constraints, when $p^{\mathcal{S}}$ and $p^{\mathcal{T}}$ have densities, the optimal transport map $G^*$ is bijective, differentiable and has a differentiable inverse, thus satisfying the change of variables formula $p^{\mathcal{S}}(\boldsymbol{x}) = p^{\mathcal{T}}(G^*(\boldsymbol{x})) |\det(\nabla G^*(\boldsymbol{x}))|$. This highly non-linear change of variables condition provides insight into why optimizing Eq. 8 directly is notoriously challenging.

# 3 METHODOLOGY

## 3.1 REGULARIZED DISTRIBUTION MATCHING DISTILLATION

Our main goal is to adapt the DMD method for the unpaired I2I between an arbitrary source distribution $p^{\mathcal{S}}$ and target distribution $p^{\mathcal{T}}$. First, we note that the construction of DMD requires having only samples from the input distribution. Given this, we replace the Gaussian input $p^{\text{noise}}$ by $p^{\mathcal{S}}$, the data distribution $p^{\text{data}}$ by $p^{\mathcal{T}}$ and optimize

$$\mathcal{L}(\theta) = \int_0^T \omega_t \, \mathrm{KL}\left(p_t^{G_\theta} \,\|\, p_t^{\mathcal{T}}\right) \mathrm{d}t = \int_0^T \omega_t \, \mathbb{E}_{p^{\mathcal{S}}(\boldsymbol{x}) \mathcal{N}(\varepsilon|0,I)} \log \frac{p_t^{G_\theta}(G_\theta(\boldsymbol{x}) + \sigma_t \varepsilon)}{p_t^{\mathcal{T}}(G_\theta(\boldsymbol{x}) + \sigma_t \varepsilon)} \mathrm{d}t, \tag{9}$$

where $p_t^{G_\theta}$ and $p_t^{\mathcal{T}}$ represent, respectively, the distribution of the generator output $G_\theta(\boldsymbol{x})$ and the target distribution $p^{\mathcal{T}}$, both perturbed by the forward process up to the timestep $t$.

Optimizing the objective in Eq. 9, one obtains a generator, which takes $\boldsymbol{x} \sim p^{\mathcal{S}}$ and outputs $G_\theta(\boldsymbol{x}) \sim p^{\mathcal{T}}$, so it performs the desired transfer between the two distributions. However, there are no guarantees that the input and the output will be related. Similarly to the OT problem (Eq. 8), we fix the issue by penalizing the transport cost between them. We obtain the following objective

$$\mathcal{L}_\lambda(\theta) = \int_0^T \omega_t \, \mathrm{KL}\left(p_t^{G_\theta} \,\|\, p_t^{\mathcal{T}}\right) \mathrm{d}t + \lambda \, \mathbb{E}_{p^{\mathcal{S}}(\boldsymbol{x})} c\left(\boldsymbol{x}, G_\theta(\boldsymbol{x})\right) \to \min_\theta, \tag{10}$$

where $c(\cdot, \cdot)$ is the cost function, which describes the object properties that we aim to preserve after transfer, and $\lambda$ is the regularization coefficient. Choosing the appropriate $\lambda$ will result in finding a balance between fitting the target distribution and preserving the properties of the input.

As in DMD, we assume that the perturbed target distributions are represented by a pre-trained diffusion model $\boldsymbol{s}_t^{\mathcal{T}}$ and approximate the generator distribution score $\boldsymbol{s}_t^{G_\theta}$ by the additional fake diffusion model $\boldsymbol{s}_t^{\phi}$. Analogous to the DMD procedure (Eq. 6), we perform the coordinate descent in which, however, the generator objective is now regularized. We call the procedure *Regularized Distribution Matching Distillation* (RDMD). Formally, we optimize

$$\begin{cases} \int_0^T \omega_t \, \mathbb{E}_{\mathcal{N}(\varepsilon|0,I) p^{\mathcal{S}}(\boldsymbol{x})} \log \frac{p_t^{\phi}(G_\theta(\boldsymbol{x}) + \sigma_t \varepsilon)}{p_t^{\mathcal{T}}(G_\theta(\boldsymbol{x}) + \sigma_t \varepsilon)} \, \mathrm{d}t + \lambda \, \mathbb{E}_{p^{\mathcal{S}}(\boldsymbol{x})} c\left(\boldsymbol{x}, G_\theta(\boldsymbol{x})\right) \to \min_\theta; \\ \int_0^T \beta_t \, \mathbb{E}_{\mathcal{N}(\varepsilon|0,I) p^{\mathcal{S}}(\boldsymbol{x})} \| D_t^{\phi}(G_\theta(\boldsymbol{x}) + \sigma_t \varepsilon) - G_\theta(\boldsymbol{x}) \|^2 \, \mathrm{d}t \to \min_\phi. \end{cases} \tag{11}$$

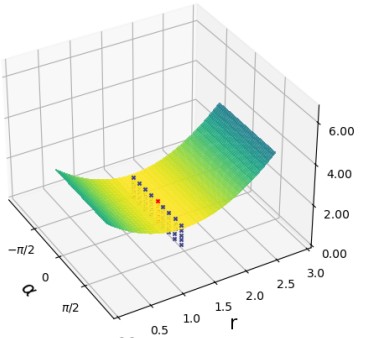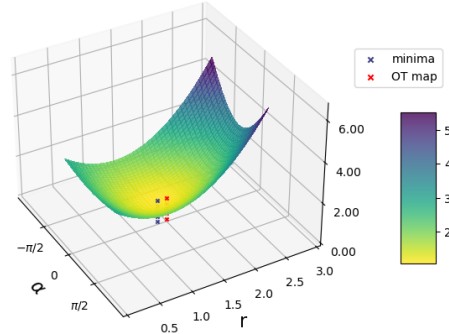

Figure 1: Comparison of the DMD loss surfaces without (left) and with (right) transport cost regularization on a toy problem of translating $\mathcal{N}(0, I)$ to $\mathcal{N}(0, 1.5^2 I)$. We set the regularization coefficient $\lambda = 0.2$. The generator is parameterized as $r \cdot C(\alpha)$, where $C(\alpha)$ is the rotation matrix, corresponding to the angle $\alpha$. Minima at the left contains all orthogonal matrices, multiplied by $\sigma = 1.5$, while the minimum at the right is attained in the only point, which is close, but not equal, to the OT map. The surfaces are moved up for the sake of visualization.

Given the optimal fake score $s_t^\phi$, the generator's objective becomes equal to the desired loss in Eq. 10, which validates the procedure.

## 3.2 ANALYSIS OF THE METHOD

The optimization problem in Eq. 10 can be seen as the soft-constrained optimal transport, which balances between satisfying the output distribution constraint and preserving the original image properties. Moreover, if one takes $\lambda \approx 0$, the objective essentially becomes equivalent to the Monge problem (Eq. 8). It can be seen by replacing the $\lambda$ coefficient before the transport cost with the $1/\lambda$ coefficient before the KL divergence. For small $\lambda$, it is almost equal to $+\infty$ whenever the generator output and the target distributions differ, making the corresponding problem hard-constrained and, therefore, equivalent to the original optimal transport problem. Based on this observation, we prove

**Theorem 1.** *Let $c(\boldsymbol{x}, \boldsymbol{y})$ be the quadratic cost [2] $\|\boldsymbol{x} - \boldsymbol{y}\|^2$ and $G^\lambda$ be the theoretical optimum in the problem 10. Then, under mild regularity conditions, it converges in probability (with respect to $p^\mathcal{S}$) to the optimal transport map $G^*$, i.e.*

$$G^\lambda \xrightarrow[\lambda \to 0]{p^\mathcal{S}} G^*. \tag{12}$$

The detailed proof can be found in Appendix A. Informally, it means that the optimal transport map can be approximated by the RDMD generator, trained on Eq. 11, given a small regularization coefficient, enough capacity of the architecture, and convergence of the optimization algorithm.

It is important to consider this result from a different perspective. It is ideologically similar to the $L_2$ regularization for over-parameterized least squares regression. The original least squares, in this case, have a manifold of solutions. At the same time, by adding $L_2$ weight penalty and taking the limit as the regularization coefficient goes to zero, one obtains a solution with the least norm based on the Moore-Penrose pseudo-inverse (Moore, 1920; Penrose, 1955). In our case, numerous maps may be optimal in the original DMD procedure, since it only requires matching the distribution at output. However, taking $\lambda \approx 0$ results in a feasible solution with almost optimal transport cost. We illustrate this by comparing the loss surface with and without regularization on a toy problem in Figure 1.

---

[2] We prove the theorem only for the quadratic case due to difficulties in analyzing minima of the Monge Problem (Eq. 8) in general cases (De Philippis & Figalli, 2014). This can be mitigated by considering the Kantorovich OT formulation (Kantorovitch, 1958), which is simpler to analyze. In practice, however, one can use any cost function of interest.

## 4 RELATED WORK

In this section, we give an overview of the existing methods for solving unpaired I2I including GANs, diffusion-based methods, and methods based on optimal transport. We also cover diffusion distillation, which is one of the core concepts in the paper.

**GANs** were the prevalent paradigm in the unpaired I2I for a long time. Among other methods, CycleGAN (Zhu et al., 2017) and the concurrent DualGAN (Yi et al., 2017), DiscoGAN (Kim et al., 2017) utilized the cycle-consistency paradigm, consisting in training the transfer network along with its inverse and optimizing the consistency term along with the adversarial loss. It gave rise to the whole family of two-sided methods, including UNIT (Liu et al., 2017) and MUNIT (Huang et al., 2018) that divide the encoding into style-space and content-space and SCAN (Li et al., 2018) that splits the procedure into coarse and fine stages. **The one-sided** GAN-based methods aim to train I2I without learning the inverse for better computational efficiency. DistanceGAN (Benaim & Wolf, 2017) achieves it by learning to preserve the distance between pairs of samples, GCGAN (Fu et al., 2019) imposes geometrical consistency constraints, and CUT (Park et al., 2020) uses the contrastive loss to maximize the patch-wise mutual information between input and output.

**Diffusion-based** I2I models mostly build on modifying the diffusion process using the source image. SDEdit (Meng et al., 2021) initializes the reverse diffusion process for target distribution with the noisy source picture instead of the pure noise to maintain similarity. Many methods guide (Ho & Salimans, 2022; Epstein et al., 2023) the target diffusion process. ILVR (Choi et al., 2021) adds the correction that enforces the current noisy sample to resemble the source. EGSDE (Zhao et al., 2022) trains a classifier between domains and encourages dissimilarity between the embeddings, corresponding to the source image and the current diffusion process state. At the same time, it enforces a small distance between their downsampled versions, which allows for a balance between faithfulness and realism. The other diffusion-based approaches include two-sided methods based on the concatenation of two diffusion models (DDIB (Su et al., 2022) and CycleDiff (Wu & De la Torre, 2023)).

**Optimal transport** (Villani et al., 2009; Peyré et al., 2019) is another useful framework for the unpaired I2I. Methods based on it usually reformulate the OT problem (Eq. 8) and its modifications as Entropic OT (EOT) (Cuturi, 2013) or Schrödinger Bridge (SB) (Föllmer, 1988) to be accessible in practice. In particular, NOT (Korotin et al., 2022), ENOT (Gushchin et al., 2024a), and NSB (Kim et al., 2023a) use the Lagrangian multipliers formulation of the distribution matching constraint, which results in simulation-based adversarial training. The other methods obtain (partially) simulation-free techniques by iteratively refining the stochastic process between two distributions. De Bortoli et al. (2021); Vargas et al. (2021) define this refinement as learning of the time-reversal with the corresponding initial distribution (source or target). The newer methods are based on Flow Matching (Lipman et al., 2022; Tong et al., 2023; Albergo & Vanden-Eijnden, 2022) and the corresponding Rectification (Liu et al., 2022; Shi et al., 2024; Liu et al., 2023) procedure. While being theoretically sound, most of these methods work well for smaller dimensions (Korotin et al., 2023) but suffer from computationally hard training in large-scale scenarios.

**Diffusion distillation** techniques are mainly divided into two families. **First** family of methods suggests using the pre-trained diffusion model as a (multi-step) noise→ image mapper and learning it. This includes optimizing the regression loss between the outputs (Salimans & Ho, 2022) or learning the integrator of the corresponding ODE (Gu et al., 2023a; Song et al., 2023; Kim et al., 2023b), including ODEs with guidance (Meng et al., 2023). **Second** family of methods considers diffusion models as a source of "knowledge" that can push an arbitrary model toward matching the distributional constraint. It is commonly formalized as optimizing the Integrated KL divergence (Luo et al., 2024; Yin et al., 2023; 2024; Nguyen & Tran, 2023) by training an additional "fake" diffusion model on the generator's output distribution. Instead of the KL divergence, one can push similarity of the corresponding scores (Zhou et al., 2024) or moments (Salimans et al., 2024). Notably, these methods do not have any specific restrictions on the model structure, which allows their wide usage (e.g. in text-to-3D (Poole et al., 2022; Wang et al., 2024)). Importantly, it allows us to push the generator towards the target distribution in the I2I setting, combined with the input-output transport cost regularization.

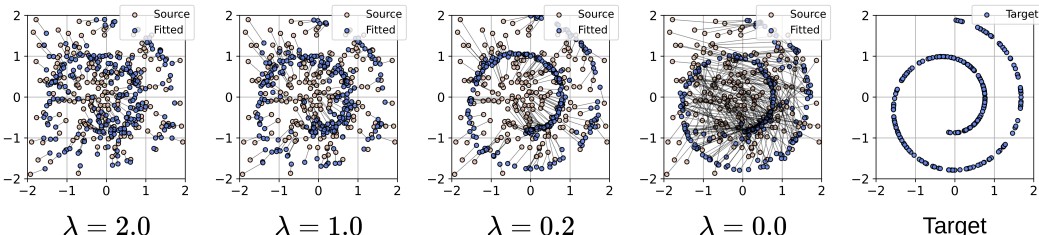

Figure 2: Visualization of RDMD mappings on *Gaussian* → *Swissroll* with different choices of the regularization coefficient $\lambda$.

## 5 EXPERIMENTS

This section presents the experimental results on several unpaired translation tasks. Section 5.1 is devoted to the toy 2D experiment. In Section 5.2 we compare our method with the diffusion-based baselines on two translation problems in $64 \times 64$ pixel space. In Section 5.3 we scale our method to $256 \times 256$ by training it in latent space of an autoencoder.

In all the experiments, we use the forward diffusion process with variance $\sigma_t = t$ and $T = 80.0$ analogous to Karras et al. (2022). We parameterize all the diffusion models with the denoiser networks $D_\sigma(\boldsymbol{x})$, conditioned on the noise level $\sigma$, and optimize Equation 3 to train the target diffusion model. As for the RDMD procedure, we optimize Equation 11, where the gradient with respect to the generator parameters is calculated analogously to Equation 7. The transport cost $c(\boldsymbol{x}, \boldsymbol{y})$ is chosen as the squared difference norm $\|\boldsymbol{x} - \boldsymbol{y}\|^2$. The average transport cost, reported in the figures, is calculated as the square root of the MSE between all input and output images for the sake of interpretability.

We use the same architecture for all networks: target score, fake score, and generator. We utilize the pre-trained target score in two ways. First, we initialize the fake model with its copy. Second, we initialize the generator $G_\theta(\boldsymbol{x})$ with the same copy $D_\sigma^{\text{real}}(\boldsymbol{x})$, but with a fixed $\sigma \in [0, T]$ (since the generator is one-step). The denoiser parameterization is trained to predict the target domain's clean images, therefore, such initialization should significantly speed up convergence and nudge the model to utilize the information about the target domain more efficiently (Nguyen & Tran, 2023; Yin et al., 2023). We explore the initialization of $\sigma$ for I2I in Appendix B. The additional training details can be found in Appendix D.

### 5.1 TOY EXPERIMENT

We validate the qualitative properties of the RDMD method on 2-dimensional *Gaussian* → *Swissroll*. In this setting, we explore the effect of varying the regularization coefficient $\lambda$ on the trained transport map $G_\theta$. In particular, we study its impact on the transport cost and fitness to the target distribution $p^{\mathcal{T}}$. In the experiment, both source and target distributions are represented with 5000 independent samples. We use the same small MLP-based architecture (Shi et al., 2024) for all the networks.

The main results are presented in Figure 2. The standard DMD ($\lambda = 0.0$) learns a transport map with several intersections when demonstrated as the set of lines between the inputs and the outputs. This observation means that the learned map is not OT, because it is not cycle-monotone (McCann, 1995). Increasing $\lambda$ yields fewer intersections, which can be used as a proxy evidence of optimality. At the same time, the generator output distribution becomes farther and farther from the desired target. The results show the importance of choosing the appropriate $\lambda$ to obtain a better trade-off between the two properties. Here, the regularization coefficient $\lambda = 0.2$ offers a good trade-off by having small intersections and producing output distribution close to the target.

### 5.2 I2I IN PIXEL SPACE

Next, we compare the proposed RDMD method with the diffusion-based baselines on the $64 \times 64$ AFHQv2 (Choi et al., 2020) *Cat* → *Wild* and CelebA (Liu et al., 2015) *Male* → *Female* translation problems. We do not compare with GAN-based methods since they mostly demonstrate results that

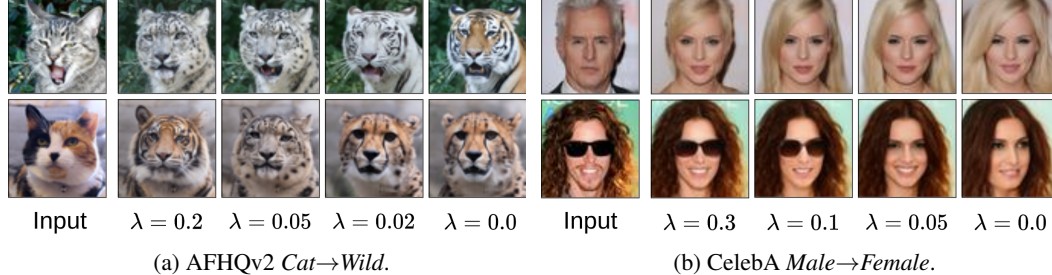

| Input | $\lambda = 0.2$ | $\lambda = 0.05$ | $\lambda = 0.02$ | $\lambda = 0.0$ | Input | $\lambda = 0.3$ | $\lambda = 0.1$ | $\lambda = 0.05$ | $\lambda = 0.0$ |

(a) AFHQv2 *Cat→Wild*.      (b) CelebA *Male→Female*.

Figure 3: Visualization of RDMD outputs with different choices of the regularization coefficient $\lambda$ on image-to-image in pixel space.

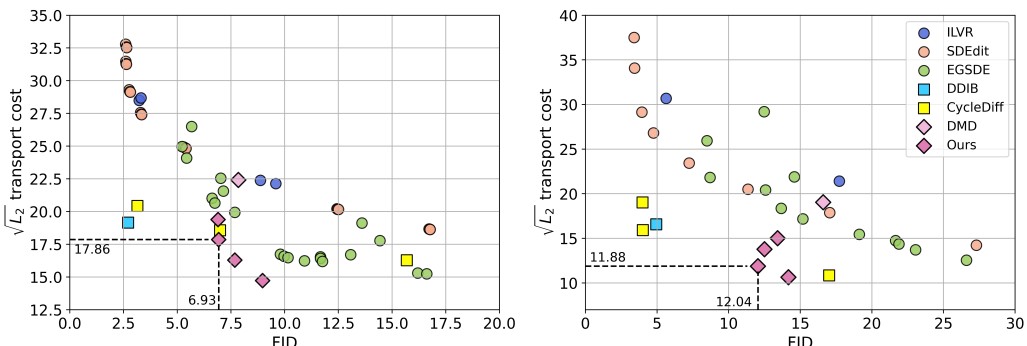

Figure 4: Comparison of RDMD with diffusion-based baselines. The figure demonstrates the tradeoff between generation quality (FID↓) and the difference between the input and output ($\sqrt{L_2}$ ↓). RDMD gives an overall better tradeoff given fairly strict requirements on the transport cost. Left: *Cat→Wild*. Right: *Male→Female*.

are inferior to those of EGSDE (Zhao et al., 2022) in terms of FID and PSNR on the same data sets with resolution $256 \times 256$.

We pre-train the target diffusion model using the DDPM++ (Song et al., 2020) architecture and EDM (Karras et al., 2022) parameterization. We slightly adapt the official baseline implementations for compatibility with the EDM setting. For each of the baselines, we run a grid of hyperparameters. The detailed hyperparameter values can be found in Appendix D.4 and D.5.

Here, we focus on investigating the faithfulness-quality trade-off achieved by our method. First, in Figure 3 we demonstrate the importance of the regularization parameter $\lambda$ in image experiments. We see that increasing $\lambda$ yields interpretable changes in model outputs (i.e. making haircut shorter or adding sunglasses), which allows for control over the model's performance. We compare the achieved faithfulness-quality trade-off with the baselines in Figure 4. The quality metric is *train* [3]FID, the faithfulness metrics are $L_2$/PSNR/SSIM. Among these metrics, we choose $L_2$ for visualization (see Figure 7 in Appendix C.1 for the full comparison in terms of PSNR and SSIM, which are, apparently, more convenient for our method).

Compared to the baselines, RDMD achieves a better trade-off given at least moderately strict requirements on the transport cost: all of our models beat the corresponding baselines in the $L_2$ range $(12.5, 17.5)$ for *Cat→Wild* and $(10.0, 15.0)$ for *Male→Female*. However, if the lower FID is strongly preferable over the transport cost, then it might be better to use one of the baselines. In this case, DDIB and CycleDiffusion show significantly better faithfulness than one-sided methods.

---

[3]We measure FID between the outputs of the model on the train source data set and the train target data set. Here, in $64 \times 64$ pixel experiments, there is not enough pictures for the test FID to be finite.

|  | FID | $\sqrt{L_2}$ | PSNR | SSIM |
|---|---|---|---|---|
| ILVR | 9.58 | 21.8 | 13.99 | 0.137 |
| SDEdit | 5.4 | 25.0 | 12.99 | 0.167 |
| EGSDE | 7.68 | 19.75 | 14.89 | 0.205 |
| DDIB | **2.71** | 19.16 | 15.25 | **0.516** |
| CycleDiff | 6.99 | 18.58 | 15.69 | 0.408 |
| DMD | 7.85 | 22.41 | 15.79 | 0.307 |
| **RDMD** | 6.93 | **17.86** | **17.84** | 0.496 |

(a) AFHQv2 *Cat → Wild*.

|  | FID | $\sqrt{L_2}$ | PSNR | SSIM |
|---|---|---|---|---|
| ILVR | 17.73 | 21.42 | 14.54 | 0.261 |
| SDEdit | 11.35 | 20.5 | 14.82 | 0.370 |
| EGSDE | 12.57 | 20.41 | 14.93 | 0.299 |
| DDIB | **4.94** | 16.58 | 16.50 | 0.597 |
| CycleDiff | 17.02 | **10.85** | 20.49 | 0.692 |
| DMD | 16.59 | 19.04 | 17.27 | 0.497 |
| **RDMD** | 12.04 | 11.88 | **20.97** | **0.701** |

(b) CelebA *Male → Female*.

Table 1: Comparison of RDMD with diffusion-based baselines on *Cat → Wild* and *Male → Female*.

|  | FID | $\sqrt{L_2}$ | LPIPS | PSNR | SSIM |
|---|---|---|---|---|---|
| ILVR | 28.85 | 118.1 | 0.557 | 11.81 | 0.326 |
| SDEdit | 28.31 | 94.00 | 0.516 | 16.69 | 0.399 |
| EGSDE | 32.26 | 72.68 | 0.466 | 16.00 | 0.430 |
| CycleDiff | 33.25 | 79.47 | 0.443 | 15.19 | 0.460 |
| DMD | 40.40 | 107.0 | 0.503 | 16.69 | 0.398 |
| **RDMD**(0.1) | 30.81 | 62.40 | **0.379** | **21.67** | **0.564** |
| EGSDE$^{\dagger}$ (p) | 30.93 | **53.44** | 0.441 | 18.32 | 0.510 |
| EGSDE (p) | 43.57 | **42.04** | 0.390 | 20.35 | 0.574 |
| **RDMD**(0.15) | 32.11 | 54.75 | **0.339** | **21.96** | **0.606** |

Table 2: Comparison of RDMD with diffusion-based baselines on $256 \times 256$ CelebA *Male → Female* in latent space. EGSDE(p) models operate in pixel space. NFE includes encoding and decoding.

For convenience, we further illustrate the comparison in Table 1 by choosing one RDMD run and comparing it with the baselines with the closest FID (i.e. we compare faithfulness given fixed realism). For both data sets, we beat almost all the baselines in terms of similarity metrics. The only exceptions are CycleDiffusion in Male→Female with better $\sqrt{L_2}$ but significantly worse FID, and DDIB with significantly lower FID but worse PSNR and $\sqrt{L_2}$. DDIB may be preferable given pre-trained diffusion models for both domains and enough resources for multi-step sampling (it requires 2 times more function evaluations than the diffusion model). If fast generation is crucial or training diffusion model for the source domain is hard, RDMD seems like a preferable method.

## 5.3 I2I IN LATENT SPACE

Finally, we demonstrate the applicability of the method in large-scale scenarios by running it in the latent space of the Stable Diffusion (Rombach et al., 2022) autoencoder. We pre-train the target diffusion model using the ADM (Dhariwal & Nichol, 2021) architecture and the EDM (Karras et al., 2022) parameterization. We run RDMD with $L_2$ transport cost and baselines in the latent space on a grid of hyperparameters, choose one RDMD run ($\lambda = 0.1$) and compare it with the baselines with the closest FID (as in Section 5.2). As previously, we compare $\sqrt{L_2}$, PSNR and SSIM as faithfulness metrics of the methods. Additionally, we compare the results with pixel-space EGSDE models (including EGSDE$^{\dagger}$) from the original paper and measure LPIPS (Zhang et al., 2018) to highlight the effect of training models in latent space [4]. We report hyperparameters and other details in Appendix D.6.

We present the results in Table 2. Qualitatively, the results are similar to the pixel space: RDMD beats the latent space diffusion-based baselines in terms of faithfulness given fixed realism [5]. Compared to the pixel-space EGSDE$^{\dagger}$ model, our method achieves worse $L_2$ distance, but wins in terms of PSNR, SSIM and LPIPS. As for the default pixel-space EGSDE, we compare it with a more faithful RDMD with $\lambda = 0.15$ and beat it in terms of FID and all similarity metrics except $L_2$. We see this

---

[4]In Table 2 FID of these models slightly differs from Table 1 by Zhao et al. (2022), since we do not additionally preprocess images. In Table 11 Zhao et al. (2022) show that this change does not violate the results.

[5]We did not include DDIB in comparison, because it unexpectedly achieved large FID = 67.36, which could indicate problems with deterministic encoding-deconding in latent space.

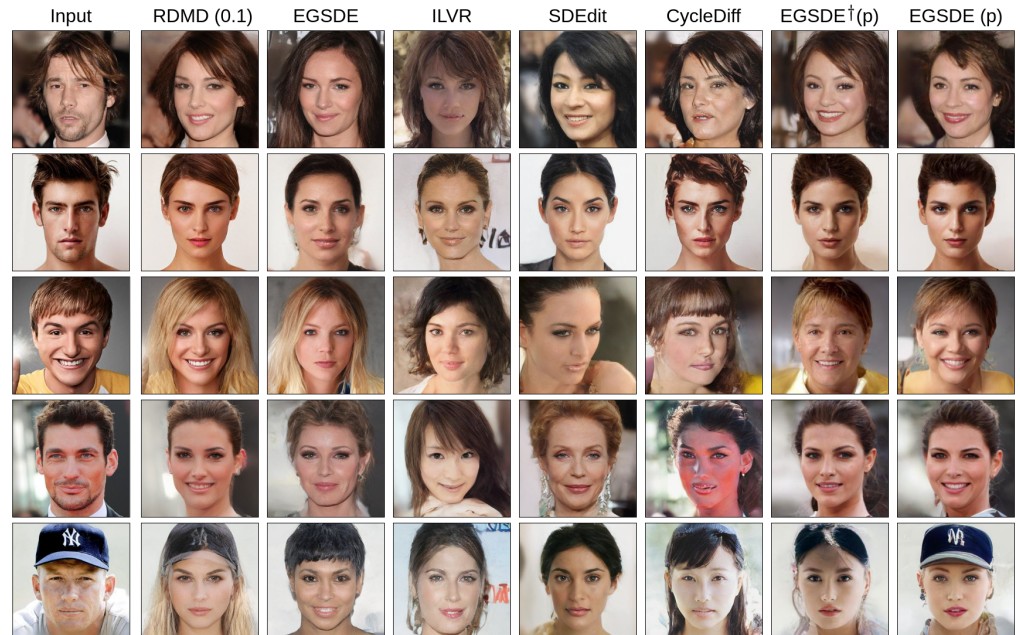

Figure 5: Comparison of RDMD in $256 \times 256$ *Male→Female* translation problem in latent space. EGSDE models, marked with "(p)", operate in the pixel space.

performance as a direct consequence of training in the latent space, which induces semantic transport cost between pictures instead of per-pixel distance. The results suggest using latent-space RDMD if one needs efficient few-step (taking into account encoding and decoding) inference and does not focus on basic per-pixel similarity. We report complete plots demonstrating the methods' tradeoff in Appendix C.2.

In Figure 5, we compare the visual performance of our method with the baselines from Table 2 on random test samples. RDMD manages to retain the original perceptual attributes and produce realistic outputs at a comparable or better level than the baselines (especially compared to the baselines in latent space) but struggles at properly translating accessories/unusual clothing components. This may suggest using different cost functions in latent space, which we leave for future work.

## 6 DISCUSSION AND LIMITATIONS

In this paper, we propose RDMD, the novel *one-step* diffusion-based algorithm for the unpaired I2I task. This algorithm is a modification of the DMD method for diffusion distillation. The main novelty is the introduction of the transport cost regularization between the input and the output of the model, which allows to control the trade-off between faithfulness and visual quality.

From the theoretical standpoint, we prove that at low regularization coefficients, the theoretical optimum of the introduced objective is close to the optimal transport map (Theorem 1). Our experiments in Section 5.1 demonstrate how the choice of regularization coefficient affects the trained mapping and allows us to build the general intuition. In Sections 5.2 and 5.3 we compare our method with the diffusion-based baselines in pixel and latent space and obtain better results given fair restrictions on the transport cost. Given fixed realism in terms of FID, our model generally achieves better faithfulness compared to the baselines, despite requiring only one function evaluation.

In terms of limitations, we admit that our theory works in the asymptotic regime, while one could derive more precise non-limit bounds. Our experimental results are limited in terms of achieving the lowest baselines' FID values (e.g. in Cat→Wild experiment we achieve 6.9, while one of the multi-step baselines, DDIB, achieves 2.71). We see making few-step modification as a potential way to mitigate this difference. Furthermore, the desired feature of the method would be switching among different regularization coefficients without re-training.

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

## APPENDIX

## A  THEORY

In this section, we aim at proving the main theoretical result of the work: solution of the soft-constrained RDMD objective converges to the solution of the hard-constrained Monge problem. Our proof is largely based on the work of Liero et al. (2018). It introduces the family of entropy-transport problems, consisting in optimizing the transport cost with soft constraints based on the divergence between the map's output distribution and the target. There are, however, differences between the problems, that prevent us from reducing the functional in Eq. 10 to the entropy-transport problems. First, authors consider the case of finite non-negative measures, while we stick to the probability distributions. Second, the family of Csiszár $f$-divergences (Csiszár, 1967), used by Liero et al. (2018), seemingly does not contain the integral ensemble of KL divergences, used in Eq. 10. Finally, we illustrate the proof in a simpler particular setting for the narrative purposes. Nevertheless, the used ideas are very similar.

## A.1 PROOF OUTLINE

We start by giving a simple outline of the proof. Given a pair of source and target distributions $p^{\mathcal{S}}$ and $p^{\mathcal{T}}$, RDMD optimizes the following functional with respect to the generator $G$:

$$\int_0^T \omega_t \, \mathrm{KL}\left(p_t^G \,\|\, p_t^{\mathcal{T}}\right) \mathrm{d}t + \lambda \, \mathbb{E}_{p^{\mathcal{S}}(\boldsymbol{x})} c\left(\boldsymbol{x}, G(\boldsymbol{x})\right), \tag{13}$$

where $p_t^G$ and $p_t^{\mathcal{T}}$ are the generator distribution $p^G$ and the target distribution $p^{\mathcal{T}}$, perturbed by the forward diffusion process up to the time step $t$. Our goal is to prove that the optimal generator of the regularized objective converges to the optimal transport map when $\lambda \to 0$. With a slight abuse of notation, in this section we will use a different objective

$$\mathcal{L}^{\alpha}(G) = \alpha \int_0^T \omega_t \, \mathrm{KL}\left(p_t^G \,\|\, p_t^{\mathcal{T}}\right) \mathrm{d}t + \mathbb{E}_{p^{\mathcal{S}}(\boldsymbol{x})} c\left(\boldsymbol{x}, G(\boldsymbol{x})\right) \tag{14}$$

and consider the equivalent limit $\alpha \to +\infty$. We also define

$$\mathcal{L}^{\infty}(G) = \begin{cases} \mathbb{E}_{p^{\mathcal{S}}(\boldsymbol{x})} c\left(\boldsymbol{x}, G(\boldsymbol{x})\right), \text{ if } p^G = p^{\mathcal{T}}; \\ +\infty, \text{ else} \end{cases} \tag{15}$$

to be the objective, corresponding to the unconditional formulation of the Monge problem (Eq. 8). In this section, we will denote minimum of this objective (which is, therefore, the optimal transport map) as $G^{\infty}$ [6]

We first assume that the infimum of the objective $\mathcal{L}^{\alpha}$ is reached and define $G^{\alpha}$ be the optimal generator. We denote by $\{\alpha_n\}_{n=1}^{+\infty}$ an arbitrary sequence with $\alpha_n \to +\infty$. We first make two informal assumptions that need to be proved (and will be in some sence further in the section):

1. The sequence $G^{\alpha_n}$ converges (in some sence) to some function $\hat{G}$;
2. $\mathcal{L}^{\alpha}$ is continuous with respect to this convergence, i.e. for every convergent sequence $G_n \to G$ holds $\mathcal{L}^{\alpha}(G_n) \to \mathcal{L}^{\alpha}(G)$.

Given this, we first observe that for each map $G$ the sequence of objectives $\mathcal{L}^{\alpha_n}(G)$ monotonically converges to the objective $\mathcal{L}^{\infty}(G)$. It follows from the fact that the first summand of $\mathcal{L}^{\alpha_n}$ converges to $+\infty$ if and only if the KL divergence is non-zero, which is equivalent to saying that $p^G$ and $p^{\mathcal{T}}$ differ (Wang et al., 2024). If instead $p^G = p^{\mathcal{T}}$, the summand zeroes out. This also means that the minimal values of the corresponding objectives form a monotonic sequence:

$$\mathcal{L}^{\alpha_n}(G^{\alpha_n}) \leq \mathcal{L}^{\alpha_{n+1}}(G^{\alpha_{n+1}}) \leq \mathcal{L}^{\infty}(G^{\infty}). \tag{16}$$

Finally, the monotonicity implies that for a fixed $m$

$$\lim_{n \to \infty} \mathcal{L}^{\alpha_n}(G^{\alpha_n}) \geq \lim_{n \to \infty} \mathcal{L}^{\alpha_m}(G^{\alpha_n}), \tag{17}$$

since the input $G^{\alpha_n}$ is fixed and $\mathcal{L}^{\alpha_n}$ monotonically increases. Using the assumed continuity of the objective, we obtain

$$\lim_{n \to \infty} \mathcal{L}^{\alpha_n}(G^{\alpha_n}) \geq \mathcal{L}^{\alpha_m}(\hat{G}) \tag{18}$$

for each $m$. Taking the limit $m \to \infty$, we obtain

$$\lim_{n \to \infty} \mathcal{L}^{\alpha_n}(G^{\alpha_n}) \geq \mathcal{L}^{\infty}(\hat{G}). \tag{19}$$

Combining this set of equations, we obtain:

$$\mathcal{L}^{\infty}(G^{\infty}) \geq \lim_{n \to \infty} \mathcal{L}^{\alpha_n}(G^{\alpha_n}) \geq \mathcal{L}^{\infty}(\hat{G}) \geq \mathcal{L}^{\infty}(G^{\infty}), \tag{20}$$

where the first inequality comes from the monotonicity of the minimal values and the last inequality uses that $G^{\infty}$ is the minimum of the objective $\mathcal{L}^{\infty}$. Hence, that limiting map $\hat{G}$ achieves minimal value of the objective $\mathcal{L}^{\infty}$ and is, therefore, the optimal transport map.

At this point, we only need to define and prove some versions of the aforementioned facts:

---

[6]Solution to the Monge problem is not always unique, but we will further impose assumptions that will guarantee the uniqueness.

1. Infimum of $\mathcal{L}^\alpha$ is reached;

2. The sequence of minima $G^{\alpha_n}$ converges;

3. $\mathcal{L}^\alpha$ is continuous with respect to this convergence.

From now on, we formulate the result in details and stick to the formal proof.

## A.2 ASSUMPTIONS AND THEOREM STATEMENT

First, we list the assumptions.

**Assumption 1.** *The distributions $p^{\mathcal{S}}$ and $p^{\mathcal{T}}$ have densities with respect to the Lebesgue measure. The distributions are defined on open bounded subsets $\mathcal{X} \subset \mathbb{R}^d$ and $\mathcal{Y} \subset \mathbb{R}^d$, where $\mathcal{Y}$ is convex. The densities are bounded away from zero and infinity on $\mathcal{X}$ and $\mathcal{Y}$, respectively.*

We admit that boundedness of the support is a very restrictive assumption from the theoretical standpoint, however in our applications (I2I) both source and target distributions are supported on the bounded space of images. We thus can set $\mathcal{X} = \mathcal{Y} = (0,1)^d$.

**Assumption 2.** *The cost $c(\boldsymbol{x}, \boldsymbol{y})$ is quadratic $\|\boldsymbol{x} - \boldsymbol{y}\|^2$.*

Here, we stick to proving the theorem only for $L_2$ cost due to difficulties in investigation of Monge map existence and regularity for general transport costs (De Philippis & Figalli, 2014).

**Assumption 3.** *The weighting function $\omega_t$ is positive and bounded.*

**Assumption 4.** *Standard deviation $\sigma_t$ of the noise, defined by the forward process, is continuous in $t$.*

**Theorem 1.** *Let $p^{\mathcal{S}}$, $p^{\mathcal{T}}$, $c$, $\omega_t$, and $\sigma_t$ satisfy the assumptions **1-3**. Then, there exists a minimum $G^\alpha$ of the objective $\mathcal{L}^\alpha$ from the Eq. 14. If $\alpha_n \to \infty$, the sequence $G^{\alpha_n}$ converges in probability (with respect to the source distribution) to the optimal transport map $G^\infty$:*

$$G^{\alpha_n} \xrightarrow[n\to\infty]{p^{\mathcal{S}}} G^\infty. \tag{21}$$

## A.3 THEORETICAL BACKGROUND

We start by listing all the results necessary for the proof. They are mostly related to the topics of measure theory (weak convergence, in particular) and optimal transport. Most of these classic facts can be found in the books (Bogachev & Ruas, 2007; Dudley, 2018). Otherwise, we make the corresponding citations.

**Definition 1.** *A sequence of probability distributions $p^n(\boldsymbol{x})$ converges weakly to the distribution $p(\boldsymbol{x})$ if for all continuous bounded test functions $\varphi \in \mathcal{C}_b(\mathbb{R}^d)$ holds*

$$\mathbb{E}_{p^n(\boldsymbol{x})}\varphi(\boldsymbol{x}) \xrightarrow[n\to\infty]{} \mathbb{E}_{p(\boldsymbol{x})}\varphi(\boldsymbol{x}). \tag{22}$$

*Notation: $p^n \xrightarrow{w} p$.*

**Definition 2.** *A function $f : \mathbb{R}^d \to \mathbb{R}$ is called lower semi-continuous (lsc), if for all $\boldsymbol{x}_n \to \boldsymbol{x}$ holds*

$$\liminf_{n\to\infty} f(\boldsymbol{x}_n) \geq f(\boldsymbol{x}). \tag{23}$$

**Theorem 2** (Portmanteau/Alexandrov). *$p^n \xrightarrow{w} p$ is equivalent to the following statement: for every lsc function $f$, bounded from below, holds*

$$\liminf_{n\to\infty} \mathbb{E}_{p^n(\boldsymbol{x})}f(\boldsymbol{x}) \geq \mathbb{E}_{p(\boldsymbol{x})}f(\boldsymbol{x}). \tag{24}$$

**Definition 3.** *A sequence of probability measures $p^n$ is called relatively compact, if for every subsequence $p^{n_k}$ there exists a weakly convergent subsequece $p^{n_{k_j}}$.*

**Definition 4.** *A sequence of probability measures $p^n$ is called tight, if for every $\varepsilon > 0$ there exists a compact set $K_\varepsilon$ such that $p^n(K_\varepsilon) \geq 1 - \varepsilon$ for all $n$.*

**Theorem 3.** *(Prokhorov) A sequence of probability measures $p^n$ is relatively compact if and only if it is tight. In particular, every weakly convergent sequence is tight.*

**Corollary 1.** *If there exists a function $\varphi(\boldsymbol{x})$ such that its sublevels $\{\boldsymbol{x} : \varphi(x) \leq r\}$ are compact and for all $n$*

$$\mathbb{E}_{p^n(\boldsymbol{x})} \varphi(x) \leq C$$

*holds with some constant $C$, then $p^n$ is tight.*

**Corollary 2.** *If a sequence $p^n$ is tight and all of its weakly convergent subsequences converge to the same measure $p$, then $p^n \xrightarrow{w} p$.*

**Definition 5.** *The functional $\mathcal{L}(p)$ is called lower semi-continuous (lsc) with respect to the weak convergence if for all weakly convergent sequences $p^n \xrightarrow{w} p$ holds*

$$\liminf_{n \to \infty} \mathcal{L}(p^n) \geq \mathcal{L}(p). \tag{25}$$

**Theorem 4** (Posner (1975)). *The KL divergence $\mathrm{KL}(p \,\|\, q)$ is lsc (in sense of weak convergence) with respect to each argument, i.e. if $p^n \xrightarrow{w} p$ and $q^n \xrightarrow{w} q$, then*

$$\liminf_{n \to \infty} \mathrm{KL}(p^n \,\|\, q) \geq \mathrm{KL}(p \,\|\, q) \tag{26}$$

$$\liminf_{n \to \infty} \mathrm{KL}(p \,\|\, q^n) \geq \mathrm{KL}(p \,\|\, q). \tag{27}$$

**Theorem 5** ( Donsker & Varadhan (1983)). *The KL divergence can be expressed as*

$$\mathrm{KL}(p\|q) = \sup_{g} \left( \mathbb{E}_{p(\boldsymbol{x})} g(\boldsymbol{x}) - \log \mathbb{E}_{q(\boldsymbol{x})} e^{g(\boldsymbol{x})} \right). \tag{28}$$

**Definition 6.** *The expression*

$$\mathbb{E}_{p(\boldsymbol{x})} e^{i \langle s, \boldsymbol{x} \rangle} \tag{29}$$

*is called the characteristic function (Fourier transform) of the distribution $p(\boldsymbol{x})$.*

**Theorem 6** (Lévy). *Weak convergence of probability measures $p^n \xrightarrow{w} p$ is equivalent to the point-wise convergence of characteristic functions, i.e. $\mathbb{E}_{p^n(\boldsymbol{x})} e^{i \langle s, \boldsymbol{x} \rangle} \to \mathbb{E}_{p(\boldsymbol{x})} e^{i \langle s, \boldsymbol{x} \rangle}$ for all $s$.*

**Definition 7.** *A sequence of measurable functions $\varphi^n(\boldsymbol{x})$ is said to converge in measure (in probability) to the function $\varphi$ with respect to the measure $p(\boldsymbol{x})$, if for all $\varepsilon > 0$ holds*

$$p \left( \{ \boldsymbol{x} : |\varphi^n(\boldsymbol{x}) - \varphi(\boldsymbol{x})| > \varepsilon \} \right) \to 0.$$

**Theorem 7** (Lebesgue). *Let $\varphi^n, \varphi$ be measurable functions such that $\|\varphi^n(\boldsymbol{x})\|, \|\varphi(\boldsymbol{x})\| \leq C$ and $\varphi^n(\boldsymbol{x}) \to \varphi(\boldsymbol{x})$ pointwise. Then $\mathbb{E}_{p(\boldsymbol{x})} \varphi^n(\boldsymbol{x}) \to \mathbb{E}_{p(\boldsymbol{x})} \varphi(\boldsymbol{x})$.*

**Lemma 1** (Fatou). *For any sequence of measurable functions $\varphi^n$ the function $\liminf_n \varphi^n$ is measurable and*

$$\int_a^b \liminf_{n \to \infty} \varphi^n(\boldsymbol{x}) \mathrm{d}\boldsymbol{x} \leq \liminf_{n \to \infty} \int_a^b \varphi^n(\boldsymbol{x}) \mathrm{d}\boldsymbol{x}. \tag{30}$$

**Theorem 8** ( Brenier (1991)). *Given the Assumption 1, there exists a unique optimal transport map that solves the Monge problem 8 for the quadratic cost.*

*Proof.* This result can be found e.g. in (De Philippis & Figalli, 2014, Theorem 3.1). ☐

**Theorem 9.** *Given the Assumption 1, the unique OT Monge map is continuous.*

*Proof.* This is a simplified version of (De Philippis & Figalli, 2014, Theorem 3.3). ☐

A.4 LOWER SEMI-CONTINUITY OF THE LOSS

Having defined all the needed terms and results, we start the proof by re-defining the objective in Eq. 14 with respect to the joint distribution $\pi$ input and output of the generator instead of the generator $G$ itself. Analogous to the Kantorovitch formulation of the optimal transport problem (Kantorovitch, 1958), for each measure $\pi$ on $\mathbb{R}^d \times \mathbb{R}^d$ (which is also called a *transport plan* or just plan) we define the corresponding fuctional as

$$\mathcal{L}^\alpha(\pi) = \alpha \int_0^T \omega_t \, \mathrm{KL} \left( \pi_{\boldsymbol{y},t} \,\|\, p_t^{\mathcal{T}} \right) \mathrm{d}t + \mathbb{E}_{\pi(\boldsymbol{x},\boldsymbol{y})} c\left(\boldsymbol{x}, \boldsymbol{y}\right), \tag{31}$$

where $\pi_{\boldsymbol{x}}$ and $\pi_{\boldsymbol{y}}$ are the corresponding projections (marginal distributions) of $\pi$ and $\pi_{\boldsymbol{y},t}$ is the perturbed $\boldsymbol{y}$-marginal distribution of $\pi$. Note that for $\pi$, corresponding to the joint distribution of $(\boldsymbol{x}, G(\boldsymbol{x}))$, $\mathcal{L}^{\alpha}(\pi)$ coincides with $\mathcal{L}^{\alpha}(G)$, defined in Eq. 14. Thus, we aim to optimize $\mathcal{L}^{\alpha}(\pi)$ with respect to such plans $\pi$, that their $\boldsymbol{x}$ marginal is equal to $p^{\mathcal{S}}$ and $\pi(\boldsymbol{y} = G(\boldsymbol{x})) = 1$ for some $G$.

**Definition 8.** *We will call a measure $\pi$ generator-based if its $\boldsymbol{x}$-marginal is equal to $p^{\mathcal{S}}$ and $\pi(\boldsymbol{y} = G(\boldsymbol{x}))$ for some function $G$.*

For the sake of cleariry, we note that the distributions $\pi_t^{\boldsymbol{y}}$ and $p_t^{\mathcal{T}}$ can be represented as $\pi^{\boldsymbol{y}} * q_t$ and $p^{\mathcal{T}} * q_t$, where $*$ is the convolution operation and $q_t = \mathcal{N}(0, \sigma_t^2 I)$. We thus rewrite the functional as

$$\mathcal{L}^{\alpha}(\pi) = \alpha \int_0^T \omega_t \, \mathrm{KL}\left(\pi_{\boldsymbol{y}} * q_t \,\|\, p^{\mathcal{T}} * q_t\right) \mathrm{d}t + \mathbb{E}_{\pi(\boldsymbol{x},\boldsymbol{y})} c\left(\boldsymbol{x}, \boldsymbol{y}\right), \tag{32}$$

Previously, we wanted to establish continuity of the objective. This may not be the case in general. Instead, we prove the following

**Lemma 2.** $\mathcal{L}^{\alpha}(\pi)$ *is lsc with respect to the weak convergence, i.e. for all weakly convergent sequences $\pi^n \xrightarrow{w} \pi$ holds*

$$\liminf_{n \to \infty} \mathcal{L}^{\alpha}(\pi^n) \geq \mathcal{L}^{\alpha}(\pi). \tag{33}$$

This result is a direct consequence of the Theorem 4 about lower semi-continuity of the KL divergence.

*Proof.* We start by proving that the projection and the convolution operation preserve weak convergence. For the first, we need to prove that for any test function $g \in \mathcal{C}_b(\mathbb{R}^d)$ holds

$$\mathbb{E}_{\pi_{\boldsymbol{y}}^n(\boldsymbol{y})} g(\boldsymbol{y}) \to \mathbb{E}_{\pi_{\boldsymbol{y}}(\boldsymbol{y})} g(\boldsymbol{y}) \tag{34}$$

given $\pi^n \xrightarrow{w} \pi$. For this, we note that the function $\varphi(\boldsymbol{x}, \boldsymbol{y}) = g(\boldsymbol{y})$ is also bounded and continuous and, thus

$$\mathbb{E}_{\pi_{\boldsymbol{y}}^n(\boldsymbol{y})} g(\boldsymbol{y}) = \mathbb{E}_{\pi^n(\boldsymbol{x},\boldsymbol{y})} \varphi(\boldsymbol{x}, \boldsymbol{y}) \to \mathbb{E}_{\pi(\boldsymbol{x},\boldsymbol{y})} \varphi(\boldsymbol{x}, \boldsymbol{y}) = \mathbb{E}_{\pi_{\boldsymbol{y}}(\boldsymbol{y})} g(\boldsymbol{y}). \tag{35}$$

Regarding the convolution, recall that $\pi_{\boldsymbol{y}}^n * q_t$ is the distribution of the sum of independent variables with corresponding distributions. Its characteristic function is equal to

$$\mathbb{E}_{\pi_{\boldsymbol{y}}^n * q_t(\boldsymbol{y}_t)} e^{i\langle s, \boldsymbol{y}_t \rangle} = \mathbb{E}_{\pi_{\boldsymbol{y}}^n(\boldsymbol{y}) q_t(\varepsilon_t)} e^{i\langle s, \boldsymbol{y} + \varepsilon_t \rangle} = \mathbb{E}_{\pi_{\boldsymbol{y}}^n(\boldsymbol{y})} e^{i\langle s, \boldsymbol{y} \rangle} \mathbb{E}_{q_t(\varepsilon_t)} e^{i\langle s, \varepsilon_t \rangle}. \tag{36}$$

Applying the Lévy's continuity theorem to $\pi_{\boldsymbol{y}}^n \xrightarrow{w} \pi_{\boldsymbol{y}}$, we take the limit and obtain

$$\mathbb{E}_{\pi_{\boldsymbol{y}}(\boldsymbol{y})} e^{i\langle s, \boldsymbol{y} \rangle} \mathbb{E}_{q_t(\varepsilon_t)} e^{i\langle s, \varepsilon_t \rangle} = \mathbb{E}_{\pi_{\boldsymbol{y}}(\boldsymbol{y}) q_t(\varepsilon_t)} e^{i\langle s, \boldsymbol{y} + \varepsilon_t \rangle} = \mathbb{E}_{\pi_{\boldsymbol{y}} * q_t(\boldsymbol{y}_t)} e^{i\langle s, \boldsymbol{y}_t \rangle}, \tag{37}$$

which implies

$$\mathbb{E}_{\pi_{\boldsymbol{y}}^n * q_t(\boldsymbol{y}_t)} e^{i\langle s, \boldsymbol{y}_t \rangle} \to \mathbb{E}_{\pi_{\boldsymbol{y}} * q_t(\boldsymbol{y}_t)} e^{i\langle s, \boldsymbol{y}_t \rangle}. \tag{38}$$

We apply the continuity theorem for the convolutions and obtain $\pi_{\boldsymbol{y}}^n * q_t \xrightarrow{w} \pi_{\boldsymbol{y}} * q_t$.

With this observation, we prove that the first term of $\mathcal{L}^{\alpha}(\pi)$ is lsc. First, we apply Lemma 1 (Fatou) and move the limit inside the integral

$$\liminf_{n \to \infty} \int_0^T \omega_t \, \mathrm{KL}\left(\pi_{\boldsymbol{y}}^n * q_t \,\|\, p^{\mathcal{T}} * q_t\right) \mathrm{d}t \geq \int_0^T \liminf_{n \to \infty} \omega_t \, \mathrm{KL}\left(\pi_{\boldsymbol{y}}^n * q_t \,\|\, p^{\mathcal{T}} * q_t\right) \mathrm{d}t. \tag{39}$$

Using the lower semi-continuity of the KL divergence (Theorem 4), we obtain

$$\int_0^T \liminf_{n \to \infty} \omega_t \, \mathrm{KL}\left(\pi_{\boldsymbol{y}}^n * q_t \,\|\, p^{\mathcal{T}} * q_t\right) \mathrm{d}t \geq \int_0^T \omega_t \, \mathrm{KL}\left(\pi_{\boldsymbol{y}} * q_t \,\|\, p^{\mathcal{T}} * q_t\right) \mathrm{d}t. \tag{40}$$

Finally, the Assumption 2 on the continuity of $c(\cdot, \cdot)$ implies its lower semi-coninuity. Theorem 2 (Portmanteau) states that

$$\liminf_{n \to \infty} \mathbb{E}_{\pi^n(\boldsymbol{x},\boldsymbol{y})} c(\boldsymbol{x}, \boldsymbol{y}) \geq \mathbb{E}_{\pi(\boldsymbol{x},\boldsymbol{y})} c(\boldsymbol{x}, \boldsymbol{y}). \tag{41}$$

Combining inequalities from Eq. 39, Eq. 40 and Eq. 41, we obtain

$$\liminf_{n \to \infty} \mathcal{L}^{\alpha}(\pi^n) \geq \mathcal{L}^{\alpha}(\pi). \tag{42}$$

$\square$

## A.5 EXISTENCE OF THE MINIMIZER

Now we aim to prove that the objective $\mathcal{L}^\alpha(\pi)$ has a minimum over generator-based plans. First, we need the following technical lemma about sublevels of the KL part of the functional.

**Lemma 3.** *Let $\{\pi^n\}_{n=1}^\infty$ be a sequence of generator-based plans that satisfy*

$$\int_0^T \omega_t \, \mathrm{KL}\left(\pi_{\boldsymbol{y},t}^n \,\|\, p_t^{\mathcal{T}}\right) \mathrm{d}t \leq C \tag{43}$$

*for some constant $C$. Then, the sequence $\{\pi^n\}_{n=1}^\infty$ is tight.*

*Proof.* We take arbitrary $\pi$ from the sequence and apply the Donsker-Varadhan representation (Theorem 5) of the KL divergence. We take the test function $g(\boldsymbol{x}) = \|x\|^2/(2\sigma_T^2)$ and obtain

$$\int_0^T \omega_t \, \mathrm{KL}\left(\pi_{\boldsymbol{y},t} \,\|\, p_t^{\mathcal{T}}\right) \mathrm{d}t \geq \int_0^T \omega_t \left(\mathbb{E}_{\pi_{\boldsymbol{y},t}(\boldsymbol{y}_t)} \frac{1}{2\sigma_T^2} \|\boldsymbol{y}_t\|^2 - \log \mathbb{E}_{p_t^{\mathcal{T}}(\boldsymbol{y}_t)} e^{\|\boldsymbol{y}_t\|^2/(2\sigma_T^2)}\right) \mathrm{d}t. \tag{44}$$

The choice of $g(\boldsymbol{x})$ is not very specific, i.e. every function that will produce finite expectations and integrals is suitable. In the right-hand side, we rewrite the expectations with repect to the original variable and noise:

$$\int_0^T \omega_t \left(\mathbb{E}_{\pi_{\boldsymbol{y}}(\boldsymbol{y})\mathcal{N}(\varepsilon|0,I)} \frac{1}{2\sigma_T^2} \|\boldsymbol{y} + \sigma_t\varepsilon\|^2 - \log \mathbb{E}_{p^{\mathcal{T}}(\boldsymbol{y})\mathcal{N}(\varepsilon|0,I)} e^{\|\boldsymbol{y}+\sigma_t\varepsilon\|^2/(2\sigma_T^2)}\right) \mathrm{d}t. \tag{45}$$

We rewrite $\|\boldsymbol{y} + \sigma_t\varepsilon\|^2$ as $\|\boldsymbol{y}\|^2 + 2\sigma_t\langle\boldsymbol{y}, \sigma_t\varepsilon\rangle + \sigma_t^2\|\varepsilon\|^2$ and note that expectation of the second term is zero. The first term is then equal to

$$\frac{1}{2\sigma_T^2} \int_0^T \omega_t \, \mathrm{d}t \cdot \mathbb{E}_{\pi_{\boldsymbol{y}}(\boldsymbol{y})} \|\boldsymbol{y}\|^2 + \frac{1}{2\sigma_T^2} \int_0^T \omega_t \sigma_t^2 \mathrm{d}t \cdot \mathbb{E}_{\mathcal{N}(\varepsilon|0,I)} \|\varepsilon\|^2. \tag{46}$$

Boundedness of $\omega_t$ (Assumption 3) implies that the first integral is finite and, say, equal to $C_1$. The second integral contains a product of bounded $\omega_t$ and continuous $\sigma_t^2$ (Assumtion 4), which is also integrable. We then denote the second summand by $C_2$ and rewrite the first summand as

$$C_1 \mathbb{E}_{\pi_{\boldsymbol{y}}(\boldsymbol{y})} \|\boldsymbol{y}\|^2 + C_2. \tag{47}$$

As for the second summand, we see that the expectation

$$\mathbb{E}_{p^{\mathcal{T}}(\boldsymbol{y})\mathcal{N}(\varepsilon|0,I)} e^{\|\boldsymbol{y}+\sigma_t\varepsilon\|^2/(2\sigma_T^2)} \tag{48}$$

with respect to $\varepsilon$ will be finite, because $\sigma_t^2/(2\sigma_T^2)$ is always less than $1/2$, which will make the exponent have negative degree. Moreover, simple calculations show that this function will be continuous with respect to $\sigma_t$ and have only quadratic terms with respect to $\boldsymbol{y}$ inside the exponent, i.e. have the form

$$e^{a(\sigma_t)\|\boldsymbol{y}-b(\sigma_t)\|^2+c(\sigma_t)} \tag{49}$$

with continuous $a, b, c$. We now want to prove that the expectation

$$\mathbb{E}_{p^{\mathcal{T}}(\boldsymbol{y})} e^{\alpha(\sigma_t)\|\boldsymbol{y}-\beta(\sigma_t)\|^2+\gamma(\sigma_t)} \tag{50}$$

will also be continuous in $t$. First, due to the boundedness of $\boldsymbol{y}$, this expectation is finite. Second, for $t_n \to t$:

$$\lim_{n\to\infty} \mathbb{E}_{p^{\mathcal{T}}(\boldsymbol{y})} e^{a(\sigma_{t_n})\|\boldsymbol{y}-b(\sigma_{t_n})\|^2+c(\sigma_{t_n})} = \tag{51}$$

$$= \mathbb{E}_{p^{\mathcal{T}}(\boldsymbol{y})} \lim_{n\to\infty} e^{a(\sigma_{t_n})\|\boldsymbol{y}-b(\sigma_{t_n})\|^2+c(\sigma_{t_n})} = \tag{52}$$

$$= \mathbb{E}_{p^{\mathcal{T}}(\boldsymbol{y})} e^{a(\sigma_t)\|\boldsymbol{y}-b(\sigma_t)\|^2+c(\sigma_t)} \tag{53}$$

due to the Theorem 7 (Lebesgue's dominated convergence). It is applicable, since $\boldsymbol{y}$ is bounded and all the functions are continuous, thus bounded in $[0, T]$.

We thus obtain that the second integral contains bounded $\omega_t$ multiplied by the logarithm of continuous function, which is always $\geq 1$ (positive exponent). This means that the whole integral is finite. Denoting it by $C_3$, we obtain

$$C_1 \mathbb{E}_{\pi_{\boldsymbol{y}}(\boldsymbol{y})} \|\boldsymbol{y}\|^2 + C_2 - C_3 \leq \int_0^T \omega_t \, \mathrm{KL}\left(\pi_{\boldsymbol{y},t} \,\|\, p_t^{\mathcal{T}}\right) \mathrm{d}t. \tag{54}$$

Combined with the condition of the lemma, we obtain

$$C_1 \mathbb{E}_{\pi_{\boldsymbol{y}}(\boldsymbol{y})} \|\boldsymbol{y}\|^2 + C_2 - C_3 \leq \int_0^T \omega_t \, \mathrm{KL}\left(\pi_{\boldsymbol{y},t} \,\|\, p_t^{\mathcal{T}}\right) \mathrm{d}t \leq C, \tag{55}$$

which implies

$$\mathbb{E}_{\pi_{\boldsymbol{y}}(\boldsymbol{y})} \|\boldsymbol{y}\|^2 \leq \frac{C + C_3 - C_2}{C_1} := C_4. \tag{56}$$

We thus obtained a uniform bound on some statistic with respect to all measures from $\{\pi^n\}$. The function $\|\boldsymbol{y}\|^2$ has compact sublevel sets $\{\|\boldsymbol{y}\|^2 \leq r\}$. Lemma 1 then states that the sequence $\pi_{\boldsymbol{y}}^n$ is tight, i.e. for all $\varepsilon > 0$ there is a compact set $K_\varepsilon$ with $\pi_{\boldsymbol{y}}^n(\boldsymbol{y} \in K_\varepsilon) \geq 1 - \varepsilon$.

Finally, marginal $\boldsymbol{x}$ distribution of each of the $\pi^n$ is $p^{\mathcal{S}}$, which is bounded (Assumption 1), i.e. there is a compact $K$ that $\pi^n(\boldsymbol{x} \in K) = 1$. Combined with the previous observation, we obtain

$$\pi^n(\boldsymbol{x} \in K, \boldsymbol{y} \in K_\varepsilon) \geq 1 - \varepsilon \tag{57}$$

for all $n$. The cartesian product $K \times K_\varepsilon$ is also compact. Theorem 3 (Prokhorov) then implies that the sequence $\pi^n$ is tight. $\qquad\square$

Now we are ready to prove the following

**Lemma 4.** *Infimum of the loss $\mathcal{L}^\alpha(\pi)$ over all generator-based transport plans $\pi$ (with $\pi_{\boldsymbol{x}} = p^{\mathcal{S}}$ and $\pi(\boldsymbol{y} = G(\boldsymbol{x}))$ for some $G$) is attained on some plan $\hat{\pi}$.*

*Proof.* We start by observing that there is at least one feasible $\pi$ with the aforementioned properties. For this purpose one can take the optimal transport map $G^\infty$ between $p^{\mathcal{S}}$ and $p^{\mathcal{T}}$, which is unique by Theorem 8 under Assumptions 1, 2.

Let $\pi^n$ be a sequence of feasible generator-based measures that $\mathcal{L}^\alpha(\pi^n)$ converges to the corresponding infimum $\mathcal{L}_{\mathrm{inf}}^\alpha$ (it exists by the definition of the infimum). Without loss of generality, we can assume that $\mathcal{L}^\alpha(\pi^n) \leq \mathcal{L}_{\mathrm{inf}}^\alpha + 1$ for all $n$ (if not, one can drop large enough sequence prefix). This implies that for all $n$ holds

$$\alpha \int_0^T \omega_t \, \mathrm{KL}\left(\pi_{\boldsymbol{y},t} \,\|\, p_t^{\mathcal{T}}\right) \mathrm{d}t \leq \mathcal{L}_{\mathrm{inf}}^\alpha + 1. \tag{58}$$

Lemma 3 implies that the sequence $\pi^n$ is tight. Prokhorov theorem then states that $\pi^n$ has a weakly convergent subsequence $\pi^{n_k} \xrightarrow{w} \hat{\pi}$. Lower semi-continuity of the loss $\mathcal{L}^\alpha$ implies that

$$\liminf_{k \to \infty} \mathcal{L}^\alpha(\pi^{n_k}) \geq \mathcal{L}^\alpha(\hat{\pi}) \geq \mathcal{L}_{\mathrm{inf}}^\alpha. \tag{59}$$

At the same time, $\mathcal{L}^\alpha(\pi^{n_k})$ is assumed to converge to $\mathcal{L}_{\mathrm{inf}}^\alpha$, which means that $\hat{\pi}$ is indeed the minimum. $\qquad\square$

A.6   FINISH OF THE PROOF

*Theorem 1 proof.* Finally, we combine the previous technical observations with the proof sketch from the Section A.1. Let $\alpha_n \to \infty$ be a sequence of coefficients, $G^{\alpha_n}$ be the optimal generators with respect to $\mathcal{L}^{\alpha_n}$ and $\pi^{\alpha_n}$ the joint distributions of $(\boldsymbol{x}, G^{\alpha_n}(\boldsymbol{x}))$. Additionally, we define $\pi^\infty$ to be the optimal transport plan, corresponding to $(\boldsymbol{x}, G^\infty(\boldsymbol{x}))$, where $G^\infty(\boldsymbol{x})$ is the optimal transport map. First, due to the monotonicity of $\mathcal{L}^\alpha$ with respect to $\alpha$, we have

$$\mathcal{L}^{\alpha_n}(\pi^{\alpha_n}) \leq \mathcal{L}^{\alpha_{n+1}}(\pi^{\alpha_{n+1}}) \leq \mathcal{L}^\infty(\pi^\infty). \tag{60}$$

This implies that for all $n$ holds

$$\alpha_n \int_0^T \omega_t \, \mathrm{KL}\left(\pi^{\alpha_n}_{\boldsymbol{y},t} \,\|\, p_t^{\mathcal{T}}\right) \mathrm{d}t \leq \mathcal{L}^\infty(\pi^\infty) \Rightarrow \tag{61}$$

$$\Rightarrow \int_0^T \omega_t \, \mathrm{KL}\left(\pi^{\alpha_n}_{\boldsymbol{y},t} \,\|\, p_t^{\mathcal{T}}\right) \mathrm{d}t \leq \frac{\mathcal{L}^\infty(\pi^\infty)}{\alpha_n} \leq \frac{\mathcal{L}^\infty(\pi^\infty)}{\min_n \alpha_n}, \tag{62}$$

which is finite, since $\alpha_n \to +\infty$. One more time, we apply Lemma 3 and conclude that the sequence $\pi^{\alpha_n}$ is tight.

Let $\pi^{\alpha_{n_k}}$ be its weakly convergent subsequence: $\pi^{\alpha_{n_k}} \xrightarrow{w} \hat{\pi}$. Analogously to the Section A.1, we observe that

$$\liminf_{k \to \infty} \mathcal{L}^{\alpha_{n_k}}\left(\pi^{\alpha_{n_k}}\right) \geq \liminf_{k \to \infty} \mathcal{L}^{\alpha_{n_m}}\left(\pi^{\alpha_{n_k}}\right) \geq \mathcal{L}^{\alpha_{n_m}}\left(\hat{\pi}\right) \tag{63}$$

for any fixed $m$. The first inequality is due to the monotonicity of $\mathcal{L}^\alpha$ with respect to $\alpha$ and second is the implication of lower semi-continuity of the loss $\mathcal{L}^\alpha$ with respect to weak convergence. Taking the limit $m \to \infty$, we obtain

$$\liminf_{k \to \infty} \mathcal{L}^{\alpha_{n_k}}\left(\pi^{\alpha_{n_k}}\right) \geq \mathcal{L}^\infty(\hat{\pi}). \tag{64}$$

Combining all these observations, we obtain the following sequence of inequalities

$$\mathcal{L}^\infty(\pi^\infty) \geq \liminf_{k \to \infty} \mathcal{L}^{\alpha_{n_k}}\left(\pi^{\alpha_{n_k}}\right) \geq \mathcal{L}^\infty(\hat{\pi}) \geq \mathcal{L}^\infty(\pi^\infty), \tag{65}$$

which implies that the limiting measure $\hat{\pi}$ reaches the minimum of the objective over generator-based plans. By the uniqueness of the optimal transport map $G^\infty$ under the Assumptions 1, 2, 3, we conclude that all the convergent subsequences $\pi^{\alpha_{n_k}}$ converge to the optimal measure $\pi^\infty$. Using Corollary 2 of the Prokhorov theorem, we deduce that $\pi^{\alpha_n} \xrightarrow{w} \pi^\infty$.

Finally, we want to replace the weak convergence of $\pi^{\alpha_n}$ to $\pi^\infty$ with the convergence in probability of the generators, i.e. show

$$G^{\alpha_n} \xrightarrow{p^{\mathcal{S}}} G^\infty. \tag{66}$$

To this end, we represent the corresponding probability as the expectation of the indicator and upper bound it with a continuous function:

$$p^{\mathcal{S}}\left(\|G^{\alpha_n}(\boldsymbol{x}) - G^\infty(\boldsymbol{x})\| > \varepsilon\right) = \mathbb{E}_{p^{\mathcal{S}}(\boldsymbol{x})} I\{\|G^{\alpha_n}(\boldsymbol{x}) - G^\infty(\boldsymbol{x})\| > \varepsilon\} \tag{67}$$

$$\leq \mathbb{E}_{p^{\mathcal{S}}(\boldsymbol{x})} d\left(G^{\alpha_n}(\boldsymbol{x}), G^\infty(\boldsymbol{x})\right), \tag{68}$$

where $d$ is a continuous indicator approximation, defined as

$$d(\boldsymbol{u}, \boldsymbol{v}) = \begin{cases} \frac{\|\boldsymbol{u} - \boldsymbol{v}\|}{\varepsilon}, & \text{if } 0 \leq \|\boldsymbol{u} - \boldsymbol{v}\| < \varepsilon; \\ 1, & \text{if } \|\boldsymbol{u} - \boldsymbol{v}\| \geq \varepsilon. \end{cases} \tag{69}$$

We define the test function

$$\varphi(\boldsymbol{x}, \boldsymbol{y}) = d\left(\boldsymbol{y}, G^\infty(\boldsymbol{x})\right) \tag{70}$$

and rewrite the upper bound as

$$\mathbb{E}_{p^{\mathcal{S}}(\boldsymbol{x})} d\left(G^{\alpha_n}(\boldsymbol{x}), G^\infty(\boldsymbol{x})\right) = \mathbb{E}_{p^{\mathcal{S}}(\boldsymbol{x})} \varphi(\boldsymbol{x}, G^{\alpha_n}(\boldsymbol{x})) = \mathbb{E}_{\pi^{\alpha_n}(\boldsymbol{x}, \boldsymbol{y})} \varphi(\boldsymbol{x}, \boldsymbol{y}). \tag{71}$$

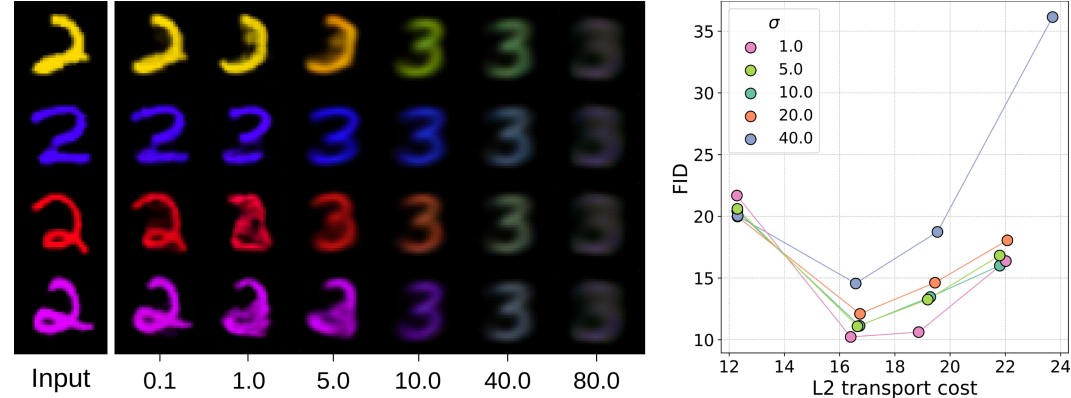

Figure 6: Left: visualization of the generator initialization at various $\sigma \in [0.1, 80.0]$, where $\sigma$ is the noise level parameter residual from the pre-trained diffusion architecture. Right: comparison of different $\sigma$ in terms of the quality-faithfulness trade-off. The metrics are obtained by initializing the generator at the corresponding $\sigma$ level and training it with the RDMD procedure. Here, $\lambda \in \{0, 1.0, 2.0, 4.0\}$. Higher $\lambda$ corresponds to the lower transport cost values.

Due to Assumptions 1, 2 and Theorem 6 the optimal transport map $G^\infty$ is continuous, which implies that this test function is bounded and continuous. Given the weak convergence of $\pi^{\alpha_n}$, we have

$$\mathbb{E}_{\pi^{\alpha_n}(\boldsymbol{x}, \boldsymbol{y})}\varphi(\boldsymbol{x}, \boldsymbol{y}) \to \mathbb{E}_{\pi^\infty(\boldsymbol{x}, \boldsymbol{y})}\varphi(\boldsymbol{x}, \boldsymbol{y}) = \mathbb{E}_{p^S(\boldsymbol{x})}\varphi(\boldsymbol{x}, G^\infty(\boldsymbol{x})) = \tag{72}$$

$$= \mathbb{E}_{p^S(\boldsymbol{x})}d(G^\infty(\boldsymbol{x}), G^\infty(\boldsymbol{x})) = 0, \tag{73}$$

which implies the desired

$$p^S\left(\|G^{\alpha_n}(\boldsymbol{x}) - G^\infty(\boldsymbol{x})\| > \varepsilon\right) \to 0. \tag{74}$$

$\square$

## B  ABLATION OF THE INITIALIZATION PARAMETER

In this section, we further explore the design space of our method by investigating the effect of the fixed generator input noise parameter $\sigma$ on the resulting quality. To this end, we take the colored version of the MNIST (LeCun, 1998) data set and perform translation between the digits "2" and "3" initializing from various $\sigma$. We use a small UNet architecture from Gushchin et al. (2024a).

The parameter $\sigma$ is residual from the pre-trained diffusion architecture and is, therefore, fixed throughout training and evaluation. However, the target denoiser network tries to convert the expected noisy input into the corresponding sample from the output distribution. Consequently, one may expect that at a suitable noise level, the generator may change the input's details to make them look appropriate for the target while preserving the original structural properties.

We demonstrate this effect on various noise levels in Figure 6. Here we observe that the small sigmas lead to the mapping close to the identity, whereas the large sigmas lead to almost constant blurry images, corresponding to the average "3" of the data set. However, there is a segment $[1.0, 10.0]$ of levels that gives a moderate-quality mapping in terms of both faithfulness and realism, which makes it a suitable initial point. Note that the FID-L2 plot is not monotone at high L2 values due to the overall poor quality of the generator, i.e. it outputs bad-quality pictures slightly related to the source. We further investigate optimal $\sigma$ choice by going through a 2D grid of the hyperparameters $(\sigma, \lambda)$ and aim to see if it is possible to choose the uniform best noise level. In Figure 6 we report the faithfulness-quality trade-off concerning various $\sigma$. We observe that there is almost monotone dependence on $\sigma$ on the segment $[1.0, 40.0]$: here the $\sigma = 1.0$ gives almost uniformly best results in terms of both metrics. Similar results are obtained by the values $5.0, 10.0$ which have fair quality visual results at initialization. Therefore, we conclude that it is best to choose the least parameter $\sigma$ among the parameters with appropriate visuals at the initial point.

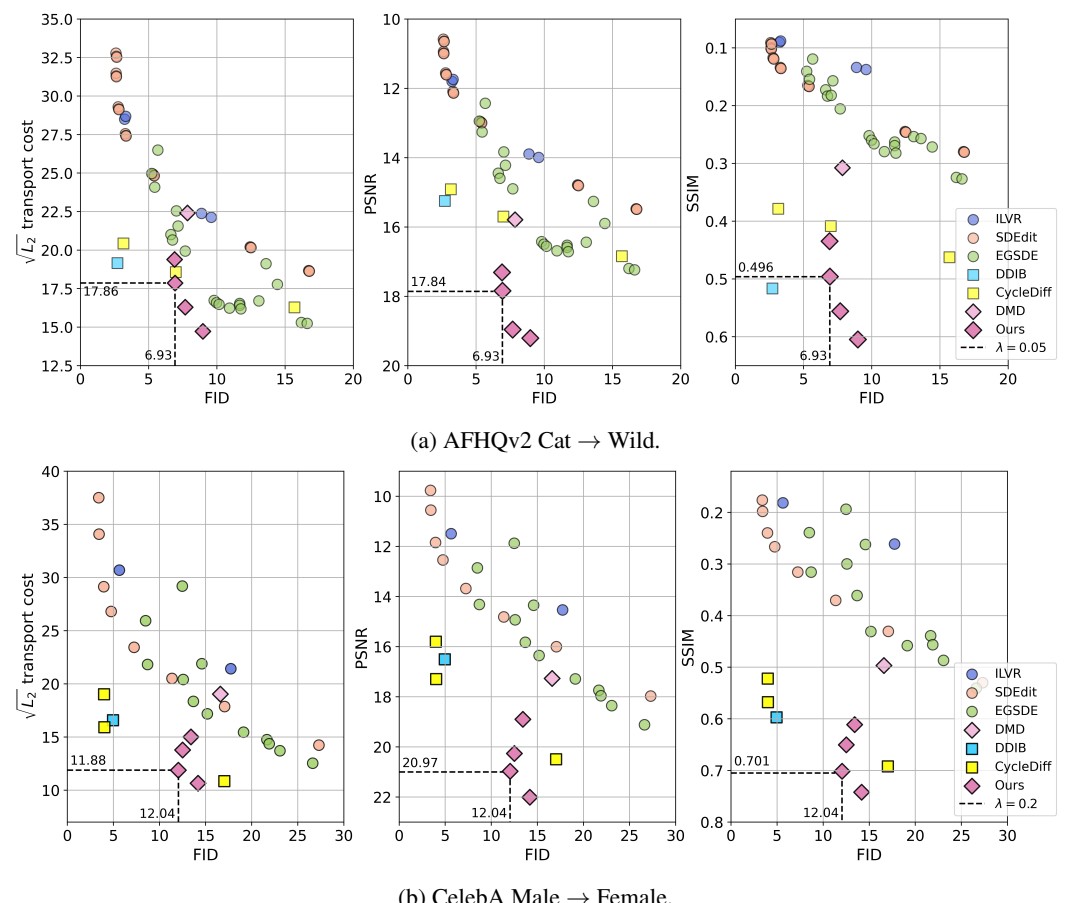

(a) AFHQv2 Cat → Wild.

(b) CelebA Male → Female.

Figure 7: Comparison of RDMD with diffusion-based baselines on $64 \times 64$ experiments in pixel space. The figure demonstrates the tradeoff between generation quality (FID↓) and the difference between the input and output (L2↓, PSNR↑, SSIM↑). RDMD gives an overall better tradeoff given fairly strict requirements on the transport cost. In the cases of PSNR and SSIM, the $y$-axis is swapped for the sake of identical readability with the first plot (left is better, low is better).

## C  ADDITIONAL COMPARISONS

### C.1  EXPERIMENTS IN PIXEL SPACE

We perform an additional visual comparison between the methods on $64 \times 64$ *Cat→Wild* and *Male→Female* in pixel space. To this end, we choose 7 pictures from the source data set and report the corresponding outputs of RDMD and the baselines in Figure 8 and Figure 9. Here, we take RDMD with $\lambda = 0.05$ for *Cat→Wild* and $\lambda = 0.3$ for *Male→Female*. As for the baselines, we choose the hyperparameters (Appendix D.4 and D.5) with the closest FID to the RDMD as it was done in Table 1.

In Section 5.2 we compare the faithfulness-realism tradeoff achieved by RDMD and the diffusion-based baselines. In Figure 4 we report tradeoff in terms of FID and $\sqrt{L_2}$ for both data sets. For the sake of completeness, in Figure 7 we report trade-off in terms of 3 faithfulness metrics: $\sqrt{L_2}$, PSNR and SSIM. Qualitatively, we still see that our method beats all the baselines given at least moderate requirements on faithfulness. Additionally, our model is strictly better than all of the one-sided baselines (ILVR, SDEdit, EGSDE) in terms of SSIM and almost strictly better than the one-sided baselines in terms of PSNR.

| Input | RDMD | EGSDE | ILVR | SDEdit | DDIB | CycleDiff |
|---|---|---|---|---|---|---|

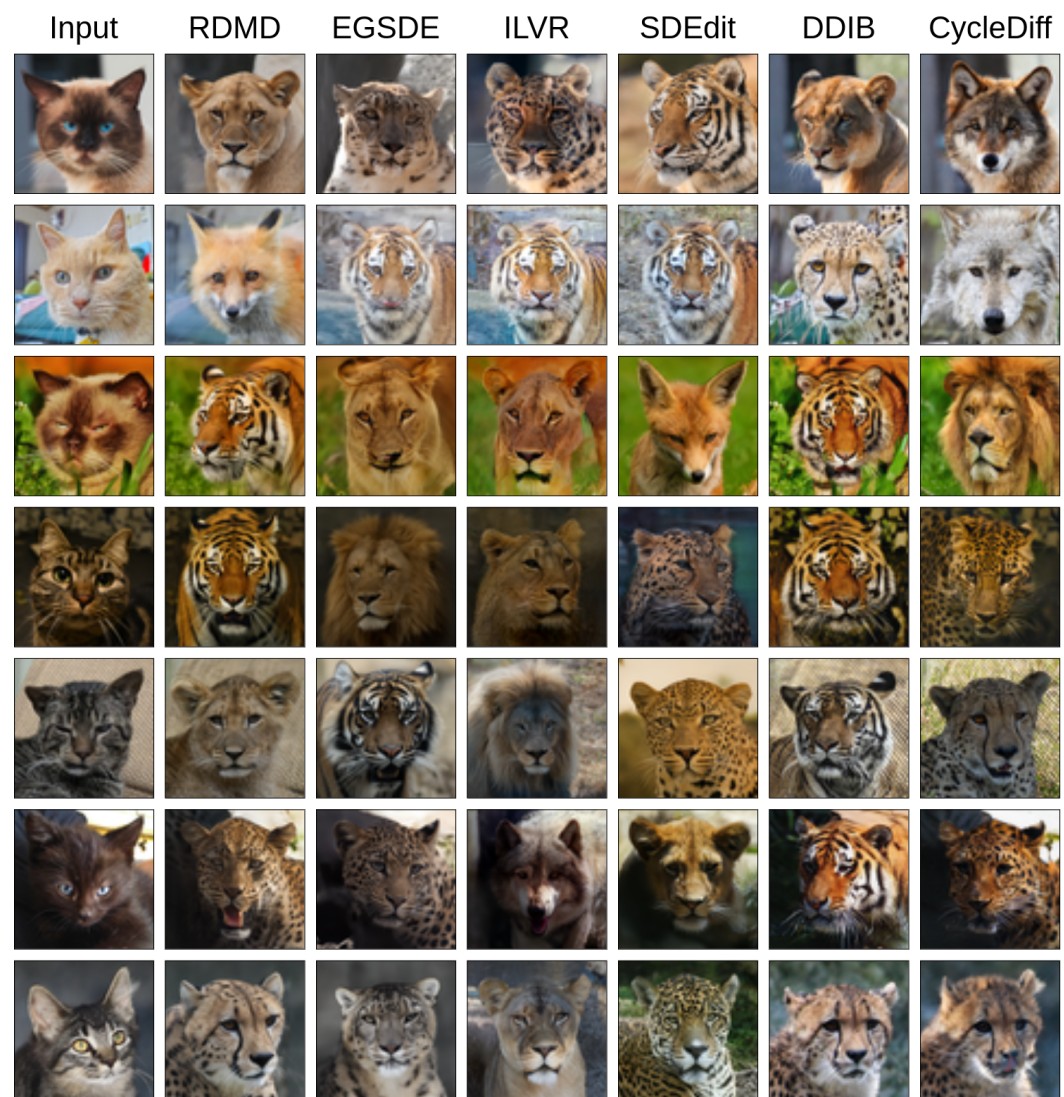

Figure 8: Visual comparison of RDMD with diffusion-based baselines on $64 \times 64$ AFHQv2 *Cat→Wild*.

## C.2 EXPERIMENTS IN LATENT SPACE

In Section 5.3 we compare the faithfulness-realism tradeoff achieved by RDMD and the diffusion-based baselines on $256 \times 256$ CelebA *Male→Female* translation task. More specifically, we choose one RDMD run and compare it to the baselines with the closest FID in terms of faithfulness metrics (i.e. compare faithfulness given fixed realism). For completeness, we report the complete comparison between all runs in Figure 10.

The results reflect pixel-space experiments: given at least moderately strict requirements on faithfulness, our model achieves a better trade-off than all of the baselines in terms of all metrics, except pixel-space EGSDE. Here, EGSDE shows comparable FID, better performance in terms of $\sqrt{L_2}$, and worse performance in all other faithfulness metrics. This is a direct consequence of the different nature of the models: latent-space for RDMD and pixel-space for EGSDE. Overall, the results suggest using RDMD if minimizing per-pixel distance is not a priority. In addition, we note that RDMD is still much more computationally efficient: it requires 3 function evaluations (encoding, translating and decoding) instead of 20+ for all diffusion-based baselines.

| Input | RDMD | EGSDE | ILVR | SDEdit | DDIB | CycleDiff |

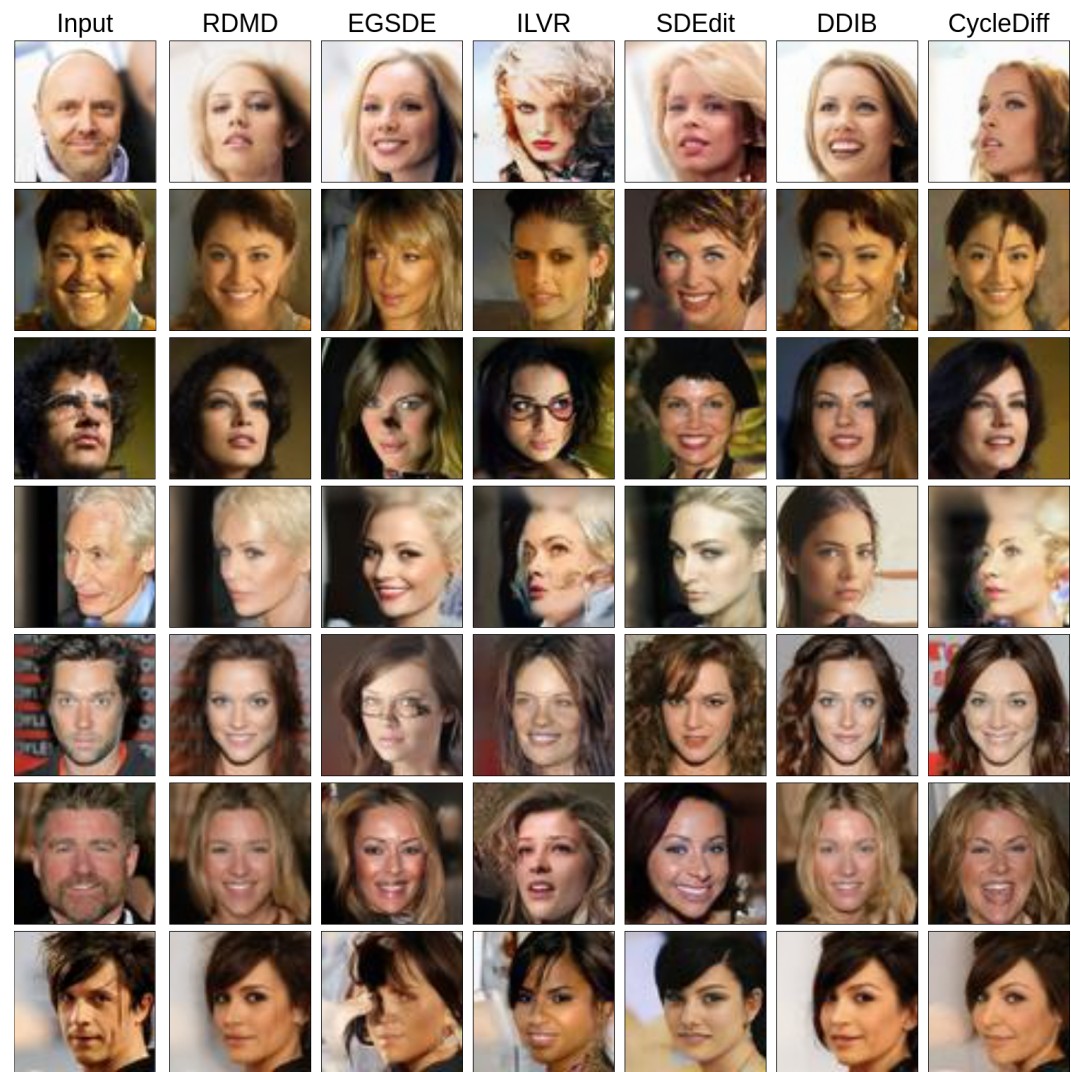

Figure 9: Visual comparison of RDMD with diffusion-based baselines on $64 \times 64$ CelebA *Male→Female* in pixel space.

## D EXPERIMENTAL DETAILS

### D.1 GENERAL DETAILS

**Metrics measurement.** In image-to-image experiments, we measure FID, $\sqrt{L_2}$ distance, PSNR, SSIM and LPIPS. We do not preprocess images before calculating the corresponding metrics (i.e. we perform measurements on images in $[0, 1]$ range with the original resolution). We use the official LPIPS (Zhang et al., 2018) implementation with VGG (Simonyan & Zisserman, 2014) backbone.

In $64 \times 64$ pixel-space experiments we measure FID between model outputs on the source train data set and the target train data set due to infinite values for the test data. In $256 \times 256$ CelebA Male→Female experiment, we measure FID between model outputs on the source test data set and the target train data set. This corresponds to the FID measurement pipeline by Park et al. (2020).

As for the transport cost $\sqrt{L_2}$, we first measure the average squared distance between inputs and outputs of the generator (without normalizing with respect to the image dimension). After averaging, we take the square root.

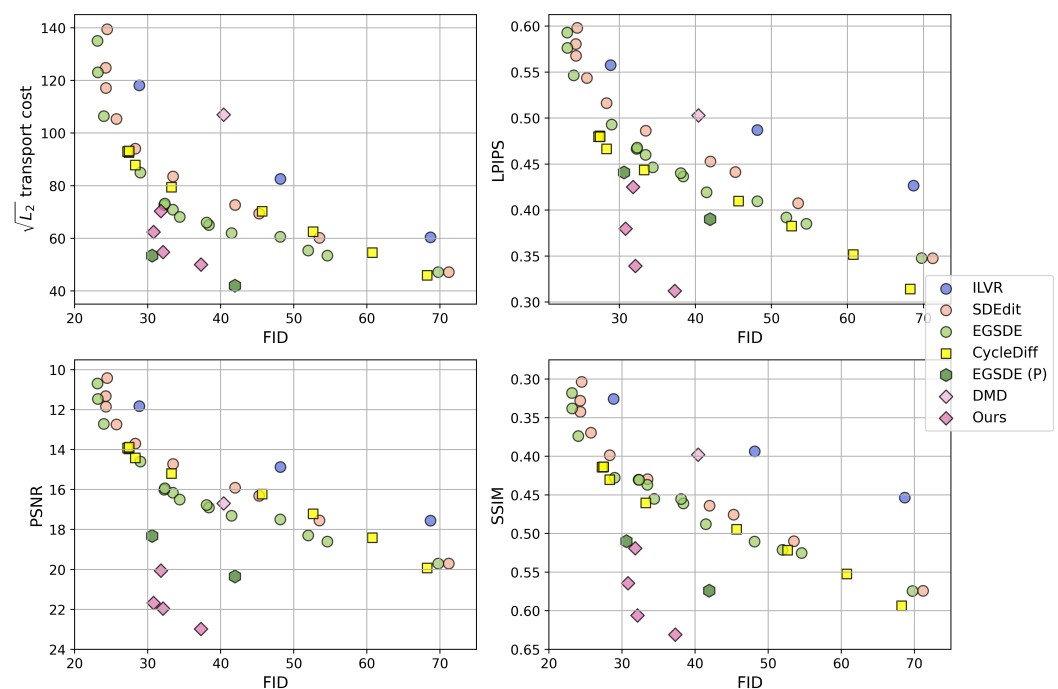

Figure 10: Comparison of RDMD with diffusion-based baselines on CelebA Male→Female in latent space. EGSDE(p) are the only baselines trained in the pixel space. The figure demonstrates the tradeoff between generation quality (FID↓) and the difference between the input and output (L2↓, LPIPS↓, PSNR↑, SSIM↑). In the cases of PSNR and SSIM, the $y$-axis is swapped for the sake of identical readability with the first plot (left is better, low is better).

## D.2 2D EXPERIMENTS

**Architecture.** We take the architecture from toy experiments of De Bortoli et al. (2021) for the diffusion model and the generator. It consists of an input-encoding MLP block, a time-encoding MLP block, and a decoding MLP block. The input-encoding MLP block consists of 4 hidden layers with dimensions $[16, 32, 32, 32]$ interspersed by LeakyReLU activations. The time-encoding MLP consists of a positional encoding layer (Vaswani et al., 2017) and follows the same MLP block structure as the input encoder. The decoding MLP block consists of 5 hidden layers with dimensions $[128, 256, 128, 64, 2]$ and operates on concatenated time embedding and input embedding each obtained from their respective encoder. The model contains $88k$ parameters.

**Training Diffusion Model.** The diffusion model is trained for 100k iterations with batch size 1024 with Adam optimizer (Kingma & Ba, 2014) with learning rate $10^{-4}$.

**Training RDMD.** Fake denoising network is trained with Adam optimizer with learning rate $10^{-4}$. The generator model is trained with a different Adam optimizer with a learning rate of $2 \cdot 10^{-5}$. We train RDMD for 100k iterations with batch size 1024.

**Computational resources.** We conduct all of the toy experiments on the CPU. Running 100k iterations with the batch size 1024 takes approximately 1 hour.

## D.3 COLORED MNIST

**Architecture.** We use the architecture from Gushchin et al. (2024a), which utilizes convolutional UNet with conditional instance normalization on time embeddings used after each upscaling block of the decoder. The model produces time embeddings via positional encoding. The model has approximately $9.9M$ parameters.

**Training Diffusion Model.** The diffusion model is trained for 24500 iterations with batch size 8192. We use the Adam optimizer with a learning rate of $4 \cdot 10^{-3}$. The model is trained in FP32. It obtains FID equal to 2.09.

**Training RDMD.** Fake denoising network is trained with Adam optimizer with a learning rate of $2 \cdot 10^{-3}$. The generator model is trained with Adam optimizer with learning rate $5 \cdot 10^{-5}$. RDMD is trained for 7300 iterations with batch size 4096.

**Computational resources.** We conduct all of the experiments on 2x NVIDIA GeForce RTX 4090 GPUs. Training Diffusion model for 24500 iterations with the batch size 8192 takes approximately 6 hours. Training RDMD for 7300 iterations with batch size 4096 takes approximately 3 hours.

### D.4 Cat2Wild

**Architecture.** We use the SongUNet architecture from EDM (Karras et al., 2022) repository, which corresponds to DDPM++ network, introduced by Song et al. (2020). The model contains approximately $55M$ parameters.

**Training Diffusion Model.** The diffusion model is trained for 80k iterations. We set the batch size to 512 and choose the best checkpoint according to FID. We use the Adam optimizer with a learning rate of $2 \cdot 10^{-4}$. We use a dropout rate equal to $0.25$ during the training and the augmentation pipeline from Karras et al. (2022) with a probability of $0.15$. The model is trained in FP32. Training takes approximately 35 hours on $4\times$ NVidia Tesla A100 80GB. The model obtains FID equal to 2.01.

**Training RDMD.** In all runs, we initialize the generator from the target diffusion model with the fixed $\sigma = 1.0$. We run 5 models, corresponding to the regularization coefficients $\lambda = \{0.0, 0.02, 0.05, 0.1, 0.2\}$. All models are trained with the Adam optimizer with a generator's learning rate of $5 \cdot 10^{-5}$ and a fake diffusion's learning rate of $3 \cdot 10^{-4}$. We train all models for 25000 iterations with batch size 512. Training takes approximately 35 hours on $4\times$ NVidia Tesla A100 80GB.

**ILVR hyperparameters.** The only hyperparameter of ILVR is the downsampling factor $N$ for the low-pass filter, which determines whether guidance would be conducted on coarser or finer information. $n_{\text{steps}}$ denotes the number of sampling steps. All metrics in Figure 7a for ILVR are obtained on the following hyperparameter grid: $N = [2, 4, 8, 16, 32]$, $n_{\text{steps}} = [18, 32, 50]$. We exclude runs with the same statistical significance and achieving FID higher than 20.0. The images in Figure 8 and the results in Table 1 (left) are obtained with hyperparameters $N = 16$ and $n_{\text{steps}} = 18$.

**SDEdit hyperparameters.** The only hyperparameter of SDEdit is the noise level $\sigma$, which acts as a starting point for sampling. The higher the noise level, the closer the sampling procedure is to unconditional generation. The smaller the noise values, the more features are carried over to the target domain at the expense of generation quality. $n_{\text{steps}}$ denotes the number of sampling steps. All metrics in Figure 7a for SDEdit are obtained on the following hyperparameter grid: $\sigma = [4, 5, 10, 15, 20, 30, 40]$, $n_{\text{steps}} = [18, 32, 50]$. We exclude runs with the same statistical significance and achieving FID higher than 20.0. The images in Figure 8 and the results in Table 1 (left) are obtained with hyperparameters $\sigma = 10$ and $n_{\text{steps}} = 50$.

**EGSDE hyperparameters.** EGSDE sampling hyperparameters include the initial noise level $\sigma$ at which the source image is perturbed, and the downsampling factor $N$ for the low-pass filter. $n_{\text{steps}}$ denotes the number of sampling steps. The method also has parameters which regulate the guidance weight of domain-specific energy term $\lambda_s$ and domain-independent energy term $\lambda_i$. We take them by default being equal to $\lambda_s = 500.0$ and $\lambda_i = 2.0$ as in the original EGSDE paper Zhao et al. (2022). All metrics in Figure 7a for EGSDE are obtained on the following hyperparameter grid: $\sigma = [5, 10, 15, 20, 40]$, $N = [8, 16, 32]$, $n_{\text{steps}} = [18, 32]$. We exclude runs with the same statistical significance and achieving FID higher than 20.0. The images in Figure 8 and the results in Table 1 (left) are obtained with hyperparameters $\sigma = 10, N = 32, n_{\text{steps}} = 50$.

**DDIB and CycleDiffusion hyperparameters.** We train an additional diffusion model with the same architecture and hyperparameters on the source domain (Cat) to further utilize it in DDIB and CycleDiffusion. The diffusion model is trained for 35k iterations. It obtains FID equal to 3.5.

We run encoding and decoding in DDIB with the deterministic EDM sampler (2nd order Heun solver) with 50 steps (100 + 100 = 200 function evaluations in total).

All metrics in Figure 4 (left) and Figure 7a for CycleDiffusion model are obtained with encoding step $T_{es} = [500, 600, 700]$ in DDPM schedule, which results in $T_{es} + T_{es}$ neural function evaluations needed for encoding the source image with the source domain network and decoding with the target domain network via DDPM ancestral sampling. The images in Figure 8 and the results in Table 1 (left) are obtained with hyperparameter $T_{es} = 600$.

### D.5 MALE2FEMALE IN PIXEL SPACE

**Architecture.** We use the SongUNet architecture from EDM (Karras et al., 2022) repository, which corresponds to DDPM++ network, introduced by Song et al. (2020). The model contains approximately $55M$ parameters.

**Training Diffusion Model.** The diffusion model is trained for 170k iterations. We set the batch size to 512 and choose the best checkpoint according to FID. We use the Adam optimizer with a learning rate of $2 \cdot 10^{-4}$. We use a dropout rate equal to 0.05 during the training and the augmentation pipeline from Karras et al. (2022) with a probability of 0.15. The model is trained in FP32. Training takes approximately 75 hours on $4\times$ NVidia Tesla A100 80GB. The model obtains FID equal to 2.65.

**Training RDMD.** In all runs, we initialize the generator from the target diffusion model with the fixed $\sigma = 1.0$. We run 5 models, corresponding to the regularization coefficients $\lambda = \{0.0, 0.05, 0.1, 0.2, 0.3\}$. All models are trained with the Adam optimizer with a generator's learning rate of $5 \cdot 10^{-5}$ and fake diffusion's learning rate of $3 \cdot 10^{-4}$. We train all models for 25000 iterations with batch size 512. Training takes approximately 35 hours on $4\times$ NVidia Tesla A100 80GB.

**ILVR hyperparameters.** The only hyperparameter of ILVR is the downsampling factor N for the low-pass filter, which determines whether guidance would be conducted on coarser or finer information. $n_{\text{steps}}$ denotes the number of sampling steps. All metrics in Figure 4 (right) and Figure 7b for ILVR are obtained on the following hyperparameter grid: N = [2, 4, 8, 16, 32], $n_{\text{steps}} = 18$. We exclude runs with the same statistical significance. For both Figure 4 (right) and Figure 7b, we include only runs with FID less than 30.0 and $\sqrt{L_2}$ transport cost lower than 40.0. The hyperparameters corresponding to results in Table 1 (right) and Figure 9 are $N = 16$.

**SDEdit hyperparameters.** The only hyperparameter of SDEdit is the noise level $\sigma$, which acts as a starting point for sampling. The higher the noise level, the closer the sampling procedure is to the unconditional generation. The smaller the noise values, the more features are carried over to the target domain at the expense of generation quality. $n_{\text{steps}}$ denotes the number of sampling steps. All metrics in Figure 4 (right) and Figure 7b for SDEdit are obtained on the following hyperparameter grid: $\sigma = [1, 2, 3, 3.4241, 5, 7, 10, 15, 20, 40, 80]$, $n_{\text{steps}} = 18$. $\sigma = 3.4241$ in EDM framework corresponds to step $T = 500$ in VP-sampling. We exclude runs with the same statistical significance. For both Figure 4 (right) and Figure 7b, we include only runs with FID less than 30.0 and $\sqrt{L_2}$ transport cost lower than 40.0. The hyperparameters corresponding to results in Table 1 (right) and Figure 9 are $\sigma = 7$.

**EGSDE hyperparameters.** EGSDE sampling hyperparameters include the initial noise level $\sigma$ at which the source image is perturbed, and the downsampling factor $N$ for the low-pass filter. $n_{\text{steps}}$ denotes the number of sampling steps. The method also has parameters which regulate the guidance weight of domain-specific energy term $\lambda_s$ and domain-independent energy term $\lambda_i$. We take them by default being equal to $\lambda_s = 500.0$ and $\lambda_i = 2.0$ as in the original EGSDE paper Zhao et al. (2022). All metrics in Figure 4 (right) and Figure 7b for EGSDE are obtained on the following hyperparameter grid: $\sigma = [3.4241, 5, 10, 20, 40]$, $N = [2, 4, 8, 16, 32]$, $n_{\text{steps}} = 18$. $\sigma = 3.4241$ in the EDM framework corresponds to step $T = 500$ in VP-sampling. We exclude runs with the same statistical significance. For both Figure 4 (right) and Figure 7b, we include only runs with FID less

than 30.0 and $\sqrt{L_2}$ transport cost lower than 40.0. The hyperparameters corresponding to results in Table 1 (right) and Figure 9 are $\sigma = 20$, $N = 16$, $n_{\text{steps}} = 18$.

**DDIB and CycleDiffusion hyperparameters.** We train an additional diffusion model with the same architecture and hyperparameters on the source domain (Male) to further utilize it in DDIB and CycleDiffusion. The diffusion model is trained for 80k iterations. It obtains FID equal to 4.11.

We run encoding and decoding in DDIB with the deterministic EDM sampler (2nd order Heun solver) with 50 steps (100 + 100 = 200 function evaluations in total).

All metrics in Figure 4 (right) and Figure 7b for CycleDiffusion model are obtained with encoding step $T_{es} = [500, 700, 1000]$ as in DDPM schedule, which results in $T_{es} + T_{es}$ neural function evaluations needed for encoding the source image with the source domain network and decoding with the target domain network via DDPM ancestral sampling. The hyperparameters corresponding to results in Table 1 (right) and Figure 9 are $T_{es} = 500$.

### D.6 MALE2FEMALE IN LATENT SPACE

**Autoencoder.** In our latent space experiments, we use the LDM-8 version of the Stable Diffusion (Rombach et al., 2022) autoencoder, which converts $256 \times 256 \times 3$ pictures into $32 \times 32 \times 4$ latent codes.

**Architecture.** We use the ADM architecture from EDM (Karras et al., 2022) repository, corresponding to the DhariwalUNet architecture (Dhariwal & Nichol, 2021), but with hyperparameters, corresponding to the LDM-8 CelebA model by Rombach et al. (2022). This includes 256 model channels, channel multipliers $[1, 2, 4]$, attention resolutions $[32, 16, 8]$ and depth 2. The model contains approximately $288M$ parameters.

**Training Diffusion Model.** The diffusion model is trained for 885k iterations. We set the batch size to 96 and choose the best checkpoint according to FID. We use the Adam optimizer with a learning rate of $1 \cdot 10^{-4}$. We use a dropout rate equal to 0.05 during training and the augmentation pipeline from Karras et al. (2022) with a probability of 0.15. The model is trained in FP32. The model is trained in FP32. Training takes approximately 130 hours on $4\times$ NVidia Tesla A100 80GB. The model obtains FID equal to 11.19.

**Training RDMD.** In all runs, we initialize the generator from the target diffusion model with the fixed $\sigma = 1.0$. We run 5 models, corresponding to the regularization coefficients $\lambda = \{0.0, 0.05, 0.10, 0.15, 0.20\}$. All models are trained with the Adam optimizer with a generator's and fake diffusion's learning rate of $2 \cdot 10^{-6}$. We train all models for 50000 iterations with batch size 96. Training takes approximately 60 hours on $2\times$NVidia Tesla A100 80GB.

**ILVR hyperparameters.** The only hyperparameter of ILVR is the downsampling factor N for the low-pass filter, which determines whether guidance would be conducted on coarser or finer information. $n_{\text{steps}}$ denotes the number of sampling steps. All metrics in Figure 10 for ILVR are obtained on the following hyperparameter grid: $N = [1.2, 1.5, 2, 4, 8, 16]$, $n_{\text{steps}} = 18$. We exclude runs with the same statistical significance. For Figure 10, we include only runs with FID less than 70.0 and $\sqrt{L_2}$ transport cost lower than 145.0. The hyperparameters corresponding to results in Table 2 and Figure 5 are $N = 16$.

**SDEdit hyperparameters.** The only hyperparameter of SDEdit is the noise level $\sigma$, which acts as a starting point for sampling. The higher the noise level, the closer the sampling procedure is to the unconditional generation. The smaller the noise values, the more features are carried over to the target domain at the expense of generation quality. $n_{\text{steps}}$ denotes the number of sampling steps. All metrics in Figure 10 for SDEdit are obtained on the following hyperparameter grid: $\sigma = [1, 2, 3, 3.4241, 5, 7, 10, 15, 20, 40]$, $n_{\text{steps}} = 18$. $\sigma = 3.4241$ in EDM framework corresponds to step $T = 500$ in VP-sampling. We exclude runs with the same statistical significance. For Figure 10, we include only runs with FID less than 70.0 and $\sqrt{L_2}$ transport cost lower than 145.0. The hyperparameters corresponding to results in Table 2 and Figure 5 are $\sigma = 7$.

**EGSDE hyperparameters.** EGSDE sampling hyperparameters include the initial noise level $\sigma$ at which the source image is perturbed, and the downsampling factor $N$ for the low-pass filter. $n_{\text{steps}}$ denotes number of sampling steps. The parameters regulating the guidance weight of domain-specific and domain-independent energy terms are denoted respectively as $\lambda_s$ and $\lambda_i$. All metrics in Figure 10 for EGSDE are obtained on the following hyperparameter grid: $\sigma = [1, 2, 3.4241, 5, 7, 10, 15, 20, 40, 60]$, $N = 4$, $n_{\text{steps}} = 18$, $\lambda_s = [80.0, 100.0]$, $\lambda_i = [0.02, 0.8, 1.5]$. $\sigma = 3.4241$ in the EDM framework corresponds to step $T = 500$ in VP-sampling. We exclude runs with the same statistical significance. For Figure 10, we include only runs with FID less than 70.0 and $\sqrt{L_2}$ transport cost lower than 145.0. The hyperparameters corresponding to results in Table 2 and Figure 5 are $\sigma = 40.0, N = 4, \lambda_s = 100, \lambda_i = 0.8$.

Pixel space EGSDE methods presented in Figure 10 and Table 2 and 5 are taken from original paper Zhao et al. (2022) with $\lambda_s = 500.0, \lambda_i = 2.0, T = 500$ for default EGSDE (p) configuration and $\lambda_s = 700.0, \lambda_i = 0.5, T = 600$ for EGSDE$^\dagger$ (p). The downsampling factor is taken as $N = 32$.

The only metric reported in Table 2 different from the ones reported by Zhao et al. (2022) is FID: authors run the evaluation pipeline by Choi et al. (2020), while we report FID without image preprocessing. We note, however, that the relative difference is small: $(30.93, 43.57)$ in Table 2 and $(30.61, 41.93)$ in Table 1 by Zhao et al. (2022). In addition, Table 11 Zhao et al. (2022) shows that changing the evaluation pipeline does not violate qualitative results. We obtain samples for visualization and measurements of FID and LPIPS by running the official implementation.

**CycleDiffusion hyperparameters.** We train an additional diffusion model with the same architecture and hyperparameters on the source domain (Male) to further utilize it in CycleDiffusion. The diffusion model is trained for 520k iterations. It obtains FID equal to 16.95.

All metrics in Figure 10 for CycleDiffusion model are obtained with encoding step $T_{es} = [200, 300, 400, 500, 600, 700, 800, 900, 1000]$ as in DDPM schedule, which results in $T_{es} + T_{es}$ neural function evaluations needed for encoding the source image with the source domain network and decoding with the target domain network via DDPM ancestral sampling. We use $T_{\text{sdedit}} = 50$ steps for additional refinement of an obtained sample with the help of the SDEdit method.

# E   COMPARISON WITH OT METHODS

In this section, we compare RDMD with the baselines that perform image-to-image translation based on solving different formulations of the optimal transport problem. Among them, OTCS (Gu et al., 2023b) defines a coupling over source and target domains and trains a conditional diffusion model between them. UOTM (Choi et al., 2024b) and DIOTM (Choi et al., 2024a) originate from OTM (Fan et al., 2021) and solve different minimax versions of the OT problem (first corresponds to the unbalanced OT formulation; second relies on insights from the dynamic formulation). Additionally, we consider currently best working models based on Schrödinger bridges: DSBM (Shi et al., 2024) and ASBM (Gushchin et al., 2024b). We exclude similar methods as NOT (Korotin et al., 2022) and OTM (Fan et al., 2021), because Choi et al. (2024b) and Choi et al. (2024) show that their performance is inferior to UOTM and DIOTM.

In Table 3, Figure 11 and Figure 12, we compare RDMD with OTCS, DSBM, UOTM and DIOTM on the $64 \times 64$ AFHQv2 *Wild→Cat* translation problem. We choose this problem because it is frequently used by the methods we compare with (besides ASBM). Since UOTM was not originally validated on I2I experiments, we take the original implementation, modify it for this scenario, and run with the default hyperparameters, suggested for larger-scale experiments. We also run the implementation of DIOTM, published in OpenReview, to measure similarity metrics and FID in (train vs train) and (test vs train) scenarios. We train OTCS on Wild->Cat with the hyperparameters used for the main unpaired I2I experiment from the paper (unpaired CelebA deblurring). We also report (test vs train) FID, measured by the authors (marked by *), which is calculated by sampling 10 output images for each source. For a more fair comparison with our one-to-one implementation of RDMD, we adapt this calculation by sampling 10 augmentations for each souce test sample (original, flipped, and 4 random crops for original and flipped) and report the obtained value as FID (text $\times 10$). We take the DSBM metrics from De Bortoli et al. (2024, Table 1).

From the quantitative results we observe that the method performs strictly better than all of the baselines, except OTCS, which has lower $L_2$, but does not fit the target distribution. RDMD thus not just provides a better faithfulness-realism trade-off, but improves on baselines in both aspects. The qualitative results in Figure 11 and Figure 12 (DSBM and DIOTM samples are taken from De Bortoli et al. (2024) and Choi et al. (2024a) respectively) confirm that RDMD generates higher-quality pictures than the baselines: OTCS does not fit the target distribution, UOTM suffers from mode collapse and unrealistic samples, DIOTM and DSBM frequently produce pictures with artifacts: distorted proportions, lack of proper facial parts etc. At the same time, we do not observe such artifacts from RDMD and obtain realistic samples that are closely related to the input.

In Table 4 and Figure 13 we compare RDMD with ASBM on the $64 \times 64$ CelebA *Male→Female* translation problem. We choose this problem as the closest $64 \times 64$ problem to the $128 \times 128$ *Male→Female* investigated by Gushchin et al. (2024b). We run the official implementation of ASBM with the hyperparameters, reported for the $128 \times 128$ problem.

Here we also observe that RDMD beats ASBM in terms of all metrics, thus offering a method with better faithfulness and realism at the same time. We validate the difference in performance in Figure 13: ASBM produces unrealistic fases with artifacts. At the same time, RDMD produces credible samples without obvious flaws. We note, however, that the RDMD samples may sometimes seem blurry. This may be caused by the optimized $L_2$ transport cost. We consider choosing a more appropriate cost function as an important future work.

|  | FID (train) | FID (test) | FID (test $\times 10$) | $\sqrt{L_2}$ | LPIPS | PSNR | SSIM |
|---|---|---|---|---|---|---|---|
| OTCS | 54.50 | 65.01 | — | **13.71** | 0.508 | 18.38 | 0.468 |
| DSBM | — | 25.41 | — | — | 0.485 | — | — |
| UOTM | 14.85 | 26.7 | — | 27.39 | 0.509 | 12.14 | 0.250 |
| DIOTM | 8.94 | 20.28 | 10.72* | 18.69 | 0.465 | 15.66 | 0.496 |
| **RDMD** | **7.87** | **18.18** | **9.24** | 15.59 | **0.363** | **19.22** | **0.594** |

Table 3: Comparison of RDMD with OT-based baselines on $64 \times 64$ AFHQv2 *Wild →Cat*. DSBM results are taken from De Bortoli et al. (2024). FID value of DIOTM, marked by *, is taken from Choi et al. (2024b), and corresponds to test vs train FID measurement with 10 generated samples per each source.

|  | FID (train) | FID (test) | $\sqrt{L_2}$ | PSNR | SSIM |
|---|---|---|---|---|---|
| ASBM | 22.94 | 31.99 | 15.32 | 17.40 | 0.524 |
| **RDMD** | **12.04** | **25.6** | **11.88** | **20.97** | **0.701** |

Table 4: Comparison of RDMD with ASBM on $64 \times 64$ CelebA *Male →Female*.

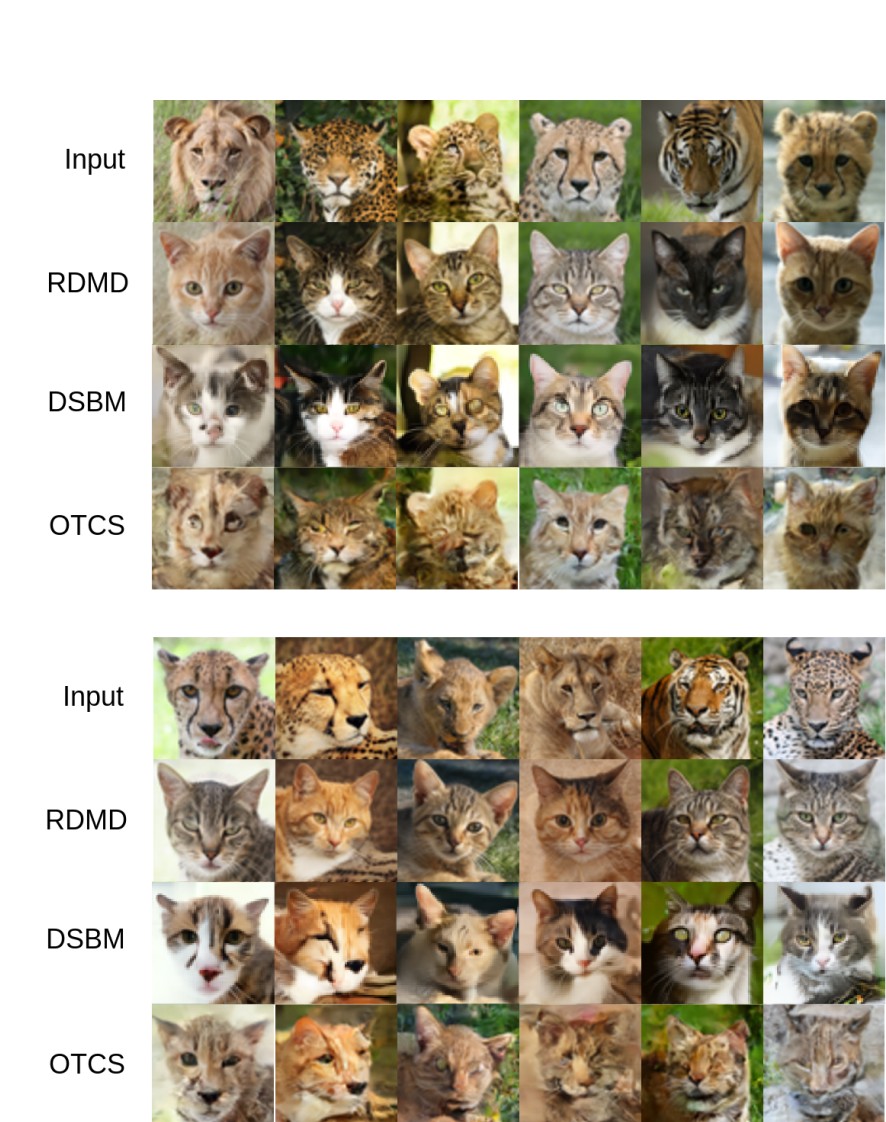

Figure 11: Visual comparison of RDMD with DSBM and OTCS on $64 \times 64$ AFHQv2 *Wild→Cat*.

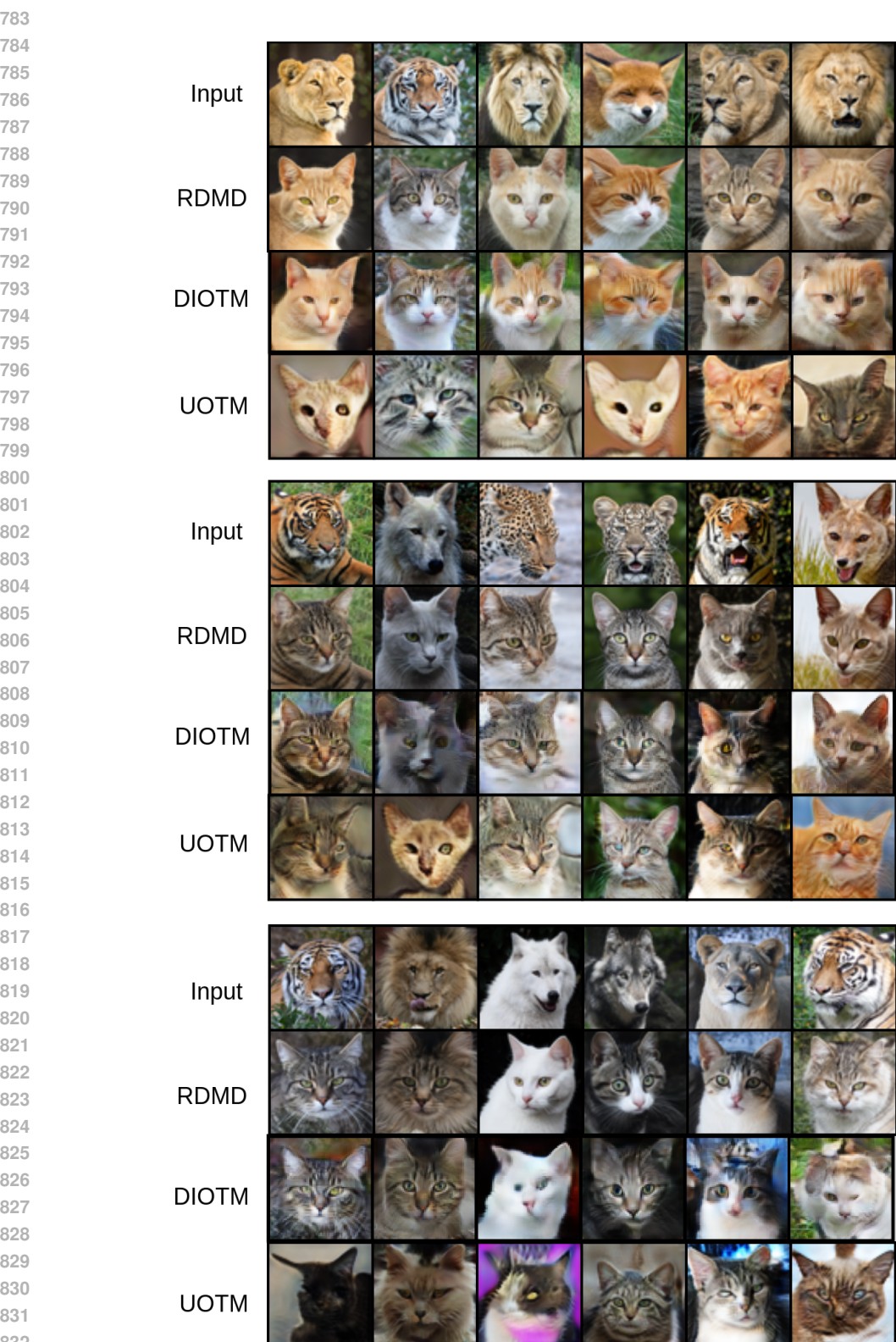

Figure 12: Visual comparison of RDMD with OT-based baselines on $64 \times 64$ AFHQv2 *Wild→Cat*.

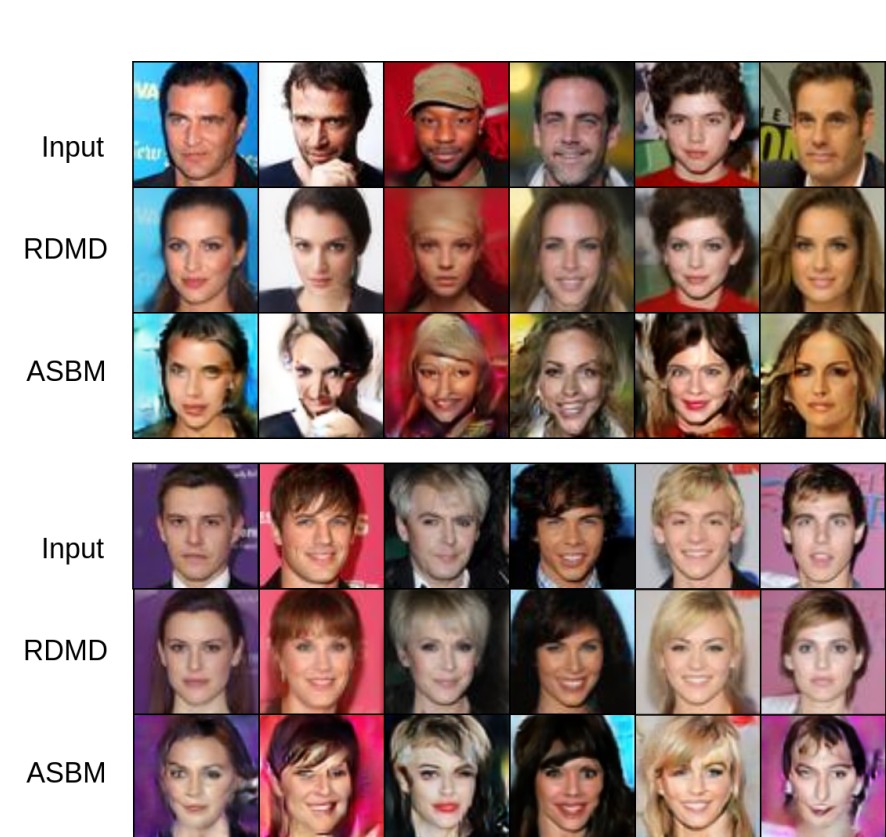

Figure 13: Visual comparison of RDMD with ASBM on $64 \times 64$ AFHQv2 *Male→Female*.

