# OpenReview forum: "Regularized Distribution Matching Distillation for One-step Unpaired Image-to-Image Translation"
_ICLR.cc/2025/Conference — Submitted to ICLR 2025_

### Official Review · Reviewer_q6cb · 2024-10-31

**Soundness:** 2
**Presentation:** 2
**Contribution:** 1
**Rating:** 3
**Confidence:** 3

**Summary:**

- This paper proposes a diffusion model distillation method for a one-step generator, called Regularized Distribution Matching Distillation (RDMD). RDMD introduces an additional cost regularizer into the previous Distribution Matching Distillation approach. The proposed method is evaluated on the image-to-image translation task, where this cost-minimization regularizer is desirable.

**Strengths:**

- This paper provides the relationship between RDMD and the optimal transport map in Thm 1.
- This paper is easy to follow.

**Weaknesses:**

- The technical novelty of the proposed method is incremental.
- The proposed method is not compared with the optimal transport models.

**Questions:**

- Could you provide details for the assumptions to guarantee the bijectivity of the Monge optimal transport problem in Line 179?
- Could you provide the quantitative results on the toy experiments in Figure 2? Evaluating the transport cost and Wasserstein distance between the generated and target distribution would strengthen the experimental results.
- In Table 1, SDEdit and DDIB achieve competitive FID results and inferior transport costs compared to RDMD. Since the transport cost and FID metric should be considered together, could you provide the translation examples for these models?
- Do all the models in Tables 1 and 2 share the same backbone networks, except for the pixel-space EGSDE?
- Could you provide a comparison with optimal transport models trained from scratch on the image-to-image translation task? [1] reported that NOT and OTM can achieve competitive FID results with the large backbone network, DDPM++. Could you clarify the advantages of RDMD over these approaches?

[1] Choi, Jaemoo, Yongxin Chen, and Jaewoong Choi. "Improving Neural Optimal Transport via Displacement Interpolation." arXiv preprint arXiv:2410.03783 (2024).

---

> ### Author Response · Authors · 2024-11-28
>
> We thank the reviewer for valuable feedback and discussion. Below, we address weaknesses and raised questions.
>
> **(1) "Could you provide details for the assumptions to guarantee the bijectivity of the Monge optimal transport problem in Line 179?"**
>
> We would like to note that the formula that is referred to does not directly relate to our work since we consider a more general soft-constrained optimal transport setting.
>
> One can make the following assumptions to guarantee the bijectivity of $G^*$:
> 1) $\mathcal{X} = \mathcal{Y} = \Omega \subset \mathbb{R}^d$ and $\Omega$ is compact. In the case of image-to-image translation, we can take $\Omega = [0, 1]^d$;
>
> 2) $p^\mathcal{S}$ and $p^{\mathcal{T}}$ have densities with respect to the Lebesgue measure, and the densities are bounded away from zero and infinity. Both $p^\mathcal{S}$ and $p^{\mathcal{T}}$ are supported on $\Omega$;
>
> 3) $c(x, y) = h(x - y)$, where $h(x - y)$ is strictly convex.
>
> From these assumptions, it follows that $G^*$ exists and has a particular form:
> $$
> G^*(x) = x - (\nabla h)^{-1} \circ \nabla \phi(x),
> $$
>
> where $\phi$ denotes Kantorovich potential. Moreover, it is invertible.
> For more detailed information, please refer to [2], Theorem 1.17 and Remark 1.20
>
> **(2) "Could you provide the quantitative results on the toy experiments in Figure 2?"**
>
> Let us denote the source and target distributions as $p^\mathcal{S}$ and $p^{\mathcal{T}}$ and $p^{G_{\theta}}$ as the distribution of the generator. $\mathcal{W}^2$ denotes squared Wasserstein Distance computed between given distributions via the `POT` library. We compute $\mathcal{W}^2$ (discrete version) between the corresponding empirical distributions, since for dimension two the difference between discrete and continuous settings is negligible. $L_2(x, G_\theta(x))$ denotes the squared pairwise $L_2$ distance between inputs and the corresponding generated images.
>
> Please see the table below for the quantitative results on Gaussian $\rightarrow$ Swiss Roll.
>
> |               | $L_2(x, G_\theta(x))$ | $\mathcal{W}^2(p^\mathcal{S}, p^{G_{\theta}})$ | $\mathcal{W}^2(p^{G_{\theta}}, p^{\mathcal{T}})$ | $\mathcal{W}^2(p^\mathcal{S}, p^{\mathcal{T}})$ |
> | ------------- | --------------------- | ---------------------------------------------- | ------------------------------------------------ | ----------------------------------------------- |
> | $\lambda=0.0$ | 4.162                 | 0.227                                          | 0.018                                            | 0.280                                           |
> | $\lambda=0.2$ | **0.247**             | **0.217**                                      | **0.032**                                        | 0.280                                           |
> | $\lambda=1.0$ | 0.114                 | 0.112                                          | 0.081                                            | 0.280                                           |
> | $\lambda=2.0$ | 0.063                 | 0.063                                          | 0.122                                            | 0.280                                           |
>
> Let us look at the $\mathcal{W}^2(p^{G_{\theta}}, p^{\mathcal{T}})$ denoting the "closeness" of the generated distribution to the target distribution. For lower coefficients of $\lambda$ we fit the target distribution better (as can be seen in the column 3 of the table) which is expected because the KL part of the loss begins to dominate. At the same time, for $\lambda$ close to zero, but not equal to zero the pairwise loss $L_2(x, G_\theta(x))$ is fairly close to $\mathcal{W}^2(p^\mathcal{S}, p^{G_{\theta}})$ (columns 1 and 2) and for larger $\lambda$ they are almost equal. For $\lambda = 0$ the pairwise distance $L_2(x, G_\theta(x))$ gets huge as is qualitatively seen in numerous intersections in Figure 2 for $\lambda = 0$. Therefore, we are interested in situation where we have both a good target fit $\mathcal{W}^2(p^{G_{\theta}}, p^{\mathcal{T}})$ and the the pairwise distance $L_2(x, G_\theta(x))$ is close to  $\mathcal{W}^2(p^\mathcal{S}, p^{G_{\theta}})$, which means that the map is close to optimal. This holds true for small $\lambda > 0$.
>
> **(3) "In Table 1, SDEdit and DDIB achieve competitive FID results and inferior transport costs compared to RDMD. Since the transport cost and FID metric should be considered together, could you provide the translation examples for these models?"**
>
> Please, see Figures 8 and 9 in Appendix for a visual comparison of our method against the baselines.
>
> **(4) Do all the models in Tables 1 and 2 share the same backbone networks, except for the pixel-space EGSDE?**
>
> Yes, all models in Tables 1 and 2 share the same backbone DDPM++ with 55M parameters. Please see Appendix sections D.4 and D.5 for further information on backbone specifics.

---

> ### Author Response · Authors · 2024-11-28
>
> **(5) "Could you provide a comparison with optimal transport models trained from scratch on the image-to-image translation task? [1] reported that NOT and OTM can achieve competitive FID results with the large backbone network, DDPM++. Could you clarify the
> advantages of RDMD over these approaches?"**
>
> We thank the reviewer for the valuable suggestion. In the rebuttal, we add a comparison with five methods suggested by the reviewers. We ran RDMD on $64\times64$ Wild $\rightarrow$ Cat translation problem to compare with the results mentioned in the DIOTM [1] paper. Here, we consider DIOTM, UOTM [3] and DSBM [4] (mentioned by the reviewer 4Uuc) as the baselines. Moreover, we compare with the OTCS [10] method (suggested by the reviewer v4Tf). We exclude OTM [5] and NOT [6] since [1] and [3] indicate their inferior performance compared to UOTM and DIOTM. We also compare our method with ASBM on CelebA $64\times64$ Male $\rightarrow$ Female translation problem.
>
> Since UOTM is not applied for I2I tasks in the original paper, we modify its official implementation for this scenario and run with the hyperparameters suggested for the larger-scale experiments. We use the official DIOTM implementation from OpenReview and train it with the default Wild $\rightarrow$ Cat hyperparameters to obtain similarity metrics. We train OTCS on Wild $\rightarrow$ Cat with the hyperparameters used for the main unpaired I2I experiment from the paper (unpaired CelebA deblurring). We train ASBM with the hyperparameters suggested for the $128 \times 128$ CelebA experiment. We do not train DSBM but report its performance measured in [7].
>
> We note that the FID (test $\times 10$) corresponds to the FID calculation protocol used in DIOTM, where the authors generate ten samples per each source (we take this result from the paper and mark it with * ). For a more fair comparison with our one-to-one implementation of RDMD, we adapt this calculation by sampling ten augmentations for each source test sample. We present the results in the table below.
>
> |      | FID (train) | FID (test) | FID (test $\times 10$) | $\sqrt{L_2}$ | LPIPS     | PSNR      | SSIM      |
> | -------- | ----------- | ---------- | ---------------------- | ------------ | --------- | --------- | --------- |
> | OTCS     | 54.50       | 65.01      | ---                    | **13.71**        | 0.508     | 18.38     | 0.468     |
> | DSBM     | ---         | 25.41      | ---                    | ---          | 0.485     | ---       | ---       |
> | UOTM     | 14.85       | 26.7       | ---                    | 27.39        | 0.509     | 12.14     | 0.250     |
> | DIOTM    | 8.94        | 20.28      | 10.72*                 | 18.69        | 0.465     | 15.66     | 0.496     |
> | **RDMD** | **7.87**    | **18.18**  | **9.24**               | **15.59**    | **0.363** | **19.22** | **0.594** |
>
> Here, we outperform all the mentioned baselines in terms of all metrics (except OTCS [10], mentioned by the reviewer v4Tf, which has a lower $L_2$, but does not fit the target distribution). We thus obtain a method which not only has a better faithfulness-realism trade-off but which is strictly better than the baselines.
>
> We stress that the main advantage of RDMD is the usage of the DMD [11] objective for optimisation. DMD, being a method of pushing an arbitrary generator towards the target distribution, proved to be one of the SOTA methods in unconditional generation. This highly motivates its usage in problems beyond unconditional generation in any case where such distribution matching objective is needed. Methods from this family proved useful in text-to-3D generation [8, 9]. In our paper and the rebuttal, we empirically demonstrate the method's superior performance compared to the diffusion-based and OT-based baselines.
>
> **References.**
>
> * [1] Improving Neural Optimal Transport via Displacement Interpolation
> * [2] Santambrogio, F. (2015). Optimal Transport for Applied Mathematicians.
> * [3] Generative Modeling through the Semi-dual Formulation of Unbalanced Optimal Transport
> * [4] Diffusion Schrödinger Bridge Matching
> * [5] Scalable computation of monge maps with general costs
> * [6] Neural Optimal Transport
> * [7] Schrödinger Bridge Flow for Unpaired Data Translation
> * [8] DreamFusion: Text-to-3D using 2D Diffusion
> * [9] ProlificDreamer: High-Fidelity and Diverse Text-to-3D Generation with Variational Score Distillation
> * [10] Optimal Transport-Guided Conditional Score-Based Diffusion Model
> * [11] One-step Diffusion with Distribution Matching Distillation

---

> > ### Comment · Reviewer_q6cb · 2024-11-30
> >
> > I appreciate the authors for the clarifications and additional experiments. However, as commented by Reviewer 4UuC and G3fX, I still have concerns about the incremental novelty of this paper. The main technical contribution of this paper is introducing the transport cost in Eq. 10. I believe that additional technical contributions or significant performance improvement are necessary for acceptance. Therefore, I will keep my original rating.

---

### Official Review · Reviewer_4UuC · 2024-11-02

**Soundness:** 1
**Presentation:** 2
**Contribution:** 1
**Rating:** 3
**Confidence:** 4

**Summary:**

This paper perform unpaired image-to-image translation through adding quadratic regularization term on the existing distribution matching distillation method.

**Strengths:**

- Experiments were conducted on various data and resolutions in both pixel-space and latent diffusion.

**Weaknesses:**

I find the contribution of this paper to be somewhat limited.

- The method mainly introduces a quadratic regularization term, $\\| x - G_\theta (x) \\|^2 $, into existing distillation approaches. This is a very simple method. Moreover, in Equations 9-11, the paper replaces the Gaussian input $p_{noise}$ with $p_{source}$, and re-train $G_\theta(x)$ with $x\sim p_{source}$. Since they no longer put noise into $G_\theta$, this approach seems more like re-training $G_\theta$ by leveraging a well-pretrained diffusion model than a distillation method. So, this method should be seen as a fully-retrained model, not a distillation method. However, the paper only compares with zero-shot diffusion-based methods or classifier-learned-based methods. I suggest authors to compare with other methods that fully train a new I2I model with the help of diffusion models. Or I suggest the authors to compare with the GAN-based methods [1] and some of the optimal-transport-based methods [2,3,4,5].

- Overall, it currently lacks a theoretical foundation and does not appear to bring substantial methodological improvements. Moreover, the comparisons are extremely weak. Even with this weak comparison, the performance is not convincing. I believe there needs big improvements in concept, literature, experimental design, and comparison groups to strengthen the paper.

**Questions:**

- In the implementation, do this method use LPIPS as a loss term in training process as Distribution Matching Distillation (DMD) does?

- CycleDiff, DDIB needs to be defined and cited in somewhere in the paper.


[1] T. Park et al., Contrastive Learning for Unpaired Image-to-Image Translation, ECCV, 2020.

[2] J. Fan et al, Scalable computation of monge maps with general costs, NeurIPS, 2021.

[3] J. Choi et al, Generative Modeling through the Semi-dual Formulation of Unbalanced Optimal Transport, NeurIPS, 2023.

[4] Y. Shi et al., Diffusion Schrodinger Bridge Matching, NeurIPS, 2023.

[5] N. Gushchin et al., Adversarial Schrodinger Bridge Matching, NeurIPS, 2024.

---

> ### Author Response · Authors · 2024-11-28
>
> **(1) "Overall, it currently lacks a theoretical foundation and does not appear to bring substantial methodological improvements. Moreover, the comparisons are extremely weak. Even with this weak comparison, the performance is not convincing. I believe there needs big improvements in concept, literature, experimental design, and comparison groups to strengthen the paper."**
>
> In terms of theoretical foundations, we verify our method by establishing its connection with the optimal transport map. In terms of methodology, we stress that DMD [1] and concurrent distribution matching methods (iDMD [2], SiD [3]) achieve SOTA in a one-step generation. This highly motivates adapting these methods for other problems in generative modelling, such as unpaired I2I, and hopefully, achieve SOTA results (e.g. DreamFusion [4] and ProlificDreamer [5] achieve remarkable results by applying distribution matching in text-to-3D).
>
> In addition, the results presented in Figures 4, 7, 10 and Tables 1, 2, suggest that the proposed method achieves better faithfulness-realism tradeoff than the multi-step baselines while being one-step.
>
> **(2) "I suggest authors to compare with other methods that fully train a new I2I model with the help of diffusion models. Or I suggest the authors to compare with the GAN-based methods [1] and some of the optimal-transport-based methods [2,3,4,5]."**
>
> We thank the reviewer for the valuable suggestion. Below, we perform the additional comparisons with most of the mentioned baselines. Specifically, we compare RDMD with DSBM [6], UOTM [7], DIOTM [8] and OTCS [9] on $64 \times 64$ Wild $\rightarrow$ Cat and with ASBM [10] on $64 \times 64$ Male $\rightarrow$ Female.  We exclude OTM [11] and CUT [12] from comparisons, since OTM was shown [7, 8] to have inferior performance than UOTM  and DIOTM, and CUT performs worse than EGSDE [13], with which we compared.
>
> Since UOTM is not applied for I2I tasks in the original paper, we modify its official implementation for this scenario and run with the hyperparameters suggested for the larger-scale experiments. We use the official DIOTM implementation from OpenReview and train it with the default Wild->Cat hyperparameters to obtain similarity metrics. We train OTCS on Wild $\rightarrow$ Cat with the hyperparameters used for the main unpaired I2I experiment from the original paper (unpaired CelebA deblurring). We train ASBM with the hyperparameters suggested for the $128 \times 128$ CelebA experiment. We do not train DSBM but report its performance measured in [14].  We obtain the following results:
>
>
> |Wild $\rightarrow$ Cat          | FID (train) | FID (test) | $\sqrt{L_2}$ | LPIPS     | PSNR      | SSIM      |
> | -------- | ----------- | ---------- | ------------ | --------- | --------- | --------- |
> | OTCS     | 54.50       | 65.01      | **13.71**        | 0.508     | 18.38     | 0.468     |
> | DSBM     | ---         | 25.41      | ---          | 0.485     | ---       | ---       |
> | UOTM     | 14.85       | 26.7       | 27.39        | 0.509     | 12.14     | 0.250     |
> | DIOTM    | 8.94        | 20.28      | 18.69        | 0.465     | 15.66     | 0.496     |
> | **RDMD** | **7.87**    | **18.18**  | **15.59**    | **0.363** | **19.22** | **0.594** |
>
>
> | Male $\rightarrow$ Female         | FID (train) | FID (test) | $\sqrt{L_2}$ | PSNR      | SSIM      |
> | -------- | ----------- | ---------- | ------------ | --------- | --------- |
> | ASBM     | 22.94       | 31.99      | 15.32        | 17.40     | 0.524     |
> | **RDMD** | **12.04**   | **25.6**   | **11.88**    | **20.97** | **0.701** |
>
> Here, we outperform all the mentioned baselines in terms of all metrics (except OTCS [9], mentioned by the reviewer v4Tf, which has a lower $L_2$, but does not fit the target distribution). We thus obtain a method, which not also has a better faithfulness-realism trade-off, but which is strictly better than both the diffusion-based and OT-based baselines. For further qualitative comparisons, please see Appendix E of the revised paper and Figures 11, 12, 13.
>
> **(3) "In the implementation, do this method use LPIPS as a loss term in training process as Distribution Matching Distillation (DMD) does?"**
>
> In DMD, using the additional LPIPS loss is possible due to the ability to generate deterministic (noise, data) pairs from the pre-trained diffusion model. Our method, in contrast, would require (source, target) pairs, which requires a paired data set or an already trained I2I model, which is why we do not use it.
>
> **(4) "CycleDiff, DDIB needs to be defined and cited in somewhere in the paper."**
>
> We cite both methods in Line 298.  We also added the methods names in the text, thanks for pointing that out.

---

> ### Author Response · Authors · 2024-11-28
>
> **References.**
>
> * [1] One-Step Diffusion with Distribution Matching Distillation
> * [2] Improved Distribution Matching Distillation for Fast Image Synthesis
> * [3] Score identity Distillation: Exponentially Fast Distillation of Pretrained Diffusion Models for One-Step Generation
> * [4] DreamFusion: Text-to-3D using 2D Diffusion
> * [5] ProlificDreamer: High-Fidelity and Diverse Text-to-3D Generation with Variational Score Distillation
> * [6] Diffusion Schrödinger Bridge Matching
> * [7] Generative Modeling through the Semi-dual Formulation of Unbalanced Optimal Transport
> * [8] Improving Neural Optimal Transport via Displacement Interpolation
> * [9] Optimal Transport-Guided Conditional Score-Based Diffusion Model
> * [10] Adversarial Schrödinger Bridge Matching
> * [11] Scalable computation of monge maps with general costs
> * [12] Contrastive Learning for Unpaired Image-to-Image Translation
> * [13] EGSDE: Unpaired Image-to-Image Translation via Energy-Guided Stochastic Differential Equations
> * [14] Schrödinger Bridge Flow for Unpaired Data Translation

---

> > ### Comment · Reviewer_4UuC · 2024-12-02
> >
> > I appreciate the authors for the clarifications and additional experiments. Moreover, I agree that the theoretical result connecting Equation 10 and OT is interesting. However, the method based on Equation 11 cannot be regarded as either a distillation of diffusion models or a true optimal transport algorithm; it is fundamentally an empirical approach. For the empirical approaches, I think the contribution should be strengthened by (1) comparing its performance against various recent empirical unpaired image-to-image translation methods (2) across various datasets. I will maintain my score.

---

### Official Review · Reviewer_G3fX · 2024-11-03

**Soundness:** 2
**Presentation:** 2
**Contribution:** 2
**Rating:** 5
**Confidence:** 4

**Summary:**

This study proposes distilling a pre-trained diffusion model into a one-step generative model to efficiently address unpaired Image-to-Image (I2I) translation. The method uses distribution matching distillation, effective in image generation, and regularizes it for unpaired I2I tasks. The proposed approach achieves good results and is significantly efficient in the sampling phase, requiring only one generation step.

**Strengths:**

- Applying distribution matching distillation (DMD) to efficiently tackle unpaired I2I translation is an interesting approach.
- Empirical results indicate that the proposed method is effective.

**Weaknesses:**

- The contribution feels incremental, as it primarily extends existing distribution matching distillation from text-to-image generation (where the prior distribution is Gaussian) to unpaired I2I by simply substituting the Gaussian prior with a source image distribution and adding the transport constraint. Please provide deeper justifications for this approach if any exist.

- While the use of a transport constraint for regularization is sound, I am concerned about the choice of the squared difference norm for adapting DMD to unpaired I2I tasks. This transport cost may be too simple and fail to capture semantic details between the generated images from the one-step generator and the source images. A more robust cost function, such as the Energy function in EGSDE, could enhance translation capacity.

- This issue may lead to output images with low L2 loss (high faithfulness) but less realism (potentially high FID), as observed in your experiments. In the experimental section, I suggest that the authors use FID computed based solely on the target distribution, rather than both the source and target as in the current design. This adjustment would better reflect the translation performance of the proposed method, given that other metrics do not adequately capture translation quality.

- Additionally, the paper lacks references to relevant work on accelerating diffusion models without additional training, such as [1, 2, 3]. There are also unpaired I2I translation methods, like [4, 5, 6], that could utilize these acceleration techniques to generate samples more efficiently, offering alternative baselines to your method. The paper could be more comprehensive if the study regarding these baselines is included.

- Although the sampling efficiency is evident with the one-step approach, the training phase requires an additional fake model. Could the authors provide an analysis of the training resources needed for the proposed framework?

- Is it challenging for both models to converge with the proposed loss function? Could the authors provide training curves for the loss functions of both the one-step generator and the fake diffusion model?

[1] Lu et al., 2022, "DPM-Solver: A Fast ODE Solver for Diffusion Probabilistic Model Sampling in Around 10 Steps", NeurIPS'22.
[2] Zhang et al., 2023, "Fast Sampling of Diffusion Models with Exponential Integrator", ICLR'23.
[3] Wizadwongsa et al., 2023, "Accelerating Guided Diffusion Sampling with Splitting Numerical Methods", ICLR'23.
[4] Lee et al., 2023, "Minimizing Trajectory Curvature of ODE-based Generative Models", ICML'23.
[5] Liu et al., 2023, "Flow Straight and Fast: Learning to Generate and Transfer Data with Rectified Flow", ICLR'23.
[6] Min et al., 2022, "EGSDE: Unpaired Image-to-Image Translation via Energy-Guided Stochastic Differential Equations", NeurIPS'22.

**Questions:**

Please refer to the Weaknesses section for my concerns and questions.

---

> ### Author Response · Authors · 2024-11-28
>
> We thank the reviewer for the valuable discussion and comments. Below, we address the aforementioned concerns.
>
> **(1) "Please provide deeper justifications for this approach if any exist."**
>
> We agree that the idea of combining transport cost with an objective that pushes the distribution towards the target previously existed in literature [1, 2]. However, Distribution Matching Distillation [3] and the successive/concurrent works (e.g. [4], [5]) achieve SOTA results in training one-step generators by propagating knowledge from diffusion models. This motivates using the technique for a wider range of problems and hopefully achieving SOTA results in tasks beyond (un-)conditional generation. For instance, DreamFusion [6] and ProlificDreamer [7] use distribution matching loss for 3D generation and achieve remarkable results in text-to-3D.
>
> **(2) "In the experimental section, I suggest that the authors use FID computed based solely on the target distribution, rather than both the source and target as in the current design. This adjustment would better reflect the translation performance of the proposed method, given that other metrics do not adequately capture translation quality."**
>
> If we understood correctly, the remark regarding FID computation relates to the footnote "We measure FID between the outputs of the model on the train source data set and the train target data set." In order to avoid misunderstandings, here we mean that FID is measured between the train target data and the generator's outputs. Here, generator takes source training samples as input. We do not mix source and target data sets for FID computation. We will reformulate the footnote more clearly.
>
> **(3) "While the use of a transport constraint for regularization is sound, I am concerned about the choice of the squared difference norm for adapting DMD to unpaired I2I tasks. This transport cost may be too simple and fail to capture semantic details between the generated images from the one-step generator and the source images. A more robust cost function, such as the Energy function in EGSDE, could enhance translation capacity. This issue may lead to output images with low L2 loss (high faithfulness) but less realism."**
>
> While we agree that choosing the appropriate cost function is important for applications and we consider it as an important future work, we would like to note that even the direct usage of the simple $L_2$ cost frequently produces competitive results (e.g. in Neural Optimal Transport [2]).
>
> Besides, recently popular Schrödinger bridge-based methods ([8, 9]) implicitly solve entropic OT with quadratic cost. We also note that in Figures 4, 7, and 10 we visualise the trade-off between FID and not only $L_2$ but also PSNR/SSIM/LPIPS and achieve results comparable to / better than the baselines (among which there is EGSDE with the aforementioned energy function).
>
> In addition, the latent-space experiment in Section 5.3 could be seen as training a map between male and female faces with transport cost equal to the distance between latent representations. Results in Table 2 and Figure 10 suggest that this type of transport cost can indeed amplify preserving semantic properties: our model achieves better LPIPS but worse $L_2$, than the pixel-space EGSDE. If LPIPS is a representative semantic metric for the task, one may use RDMD in latent space for this purpose.
>
> **(4) "Although the sampling efficiency is evident with the one-step approach, the training phase requires an additional fake model. Could the authors provide an analysis of the training resources needed for the proposed framework?"**
>
> We provide all the details about computational resources in Appendix D.

---

> ### Author Response · Authors · 2024-11-28
>
> **(5) "Additionally, the paper lacks references to relevant work on accelerating diffusion models without additional training, such as [1, 2, 3]. There are also unpaired I2I translation methods, like [4, 5, 6], that could utilize these acceleration techniques to generate samples more efficiently, offering alternative baselines to your method. The paper could be more comprehensive if the study regarding these baselines is included."**
>
> We consider combining the existing methods of unpaired I2I with acceleration techniques as a relevant yet parallel line of work, which may itself have a significant novelty and give rise to new methods. Besides, accelerating diffusion models almost always results in a decrease in quality. Given that our method already outperforms EGSDE, one could expect it to beat the accelerated EGSDE as well. As for Rectified Flow (RF) [10], we add the comparison with DSBM [8], its SDE-based analogue (proven to have better quality than RF in [8]). The other three baselines are proposed by the reviewers. We run them on the unpaired $64 \times 64$ Wild$\rightarrow$Cat translation problem. We obtain the following results:
>
> |          | FID (train) | FID (test) | $\sqrt{L_2}$ | LPIPS     | PSNR      | SSIM      |
> | -------- | ----------- | ---------- | ------------ | --------- | --------- | --------- |
> | OTCS     | 54.50       | 65.01      | **13.71**        | 0.508     | 18.38     | 0.468     |
> | DSBM     | ---         | 25.41      | ---          | 0.485     | ---       | ---       |
> | UOTM     | 14.85       | 26.7       | 27.39        | 0.509     | 12.14     | 0.250     |
> | DIOTM    | 8.94        | 20.28      | 18.69        | 0.465     | 15.66     | 0.496     |
> | **RDMD** | **7.87**    | **18.18**  | **15.59**    | **0.363** | **19.22** | **0.594** |
>
> Here, we outperform all the mentioned baselines in terms of all metrics (except OTCS [11], mentioned by the reviewer v4Tf, which has a lower $L_2$, but does not fit the target distribution). We thus obtain a method which not only has a better faithfulness-realism trade-off but which is strictly better than the suggested baselines, including DSBM. For further qualitative comparisons, please see Appendix E and Figures 11, 12, 13.
>
> As for [12], the original paper studies unconditional generation and does not include I2I experiments. At the same time, the unpaired I2I modification of the method does not seem straightforward: one should adapt the encoder to map $x_t$ to the source distribution and measure the corresponding KL divergence, which may require novel approaches in both architecture and the procedure itself.
>
> "**(6) Is it challenging for both models to converge with the proposed loss function? Could the authors provide training curves for the loss functions of both the one-step generator and the fake diffusion model?**"
>
> Due to the initialization with the pre-trained diffusion model the convergence is fast and stable. After a few hundred warm-up iterations, fake score loss monotonically converges. DMD loss quickly reaches its equilibrium and starts oscillating.

---

> > ### Author Response · Authors · 2024-11-28
> >
> > **References**
> >
> > * [1] Scalable computation of monge maps with general costs
> > * [2] Neural Optimal Transport
> > * [3] One-Step Diffusion with Distribution Matching Distillation
> > * [4] Improved Distribution Matching Distillation for Fast Image Synthesis
> > * [5] Score identity Distillation: Exponentially Fast Distillation of Pretrained Diffusion Models for One-Step Generation
> > * [6] DreamFusion: Text-to-3D using 2D Diffusion
> > * [7] ProlificDreamer: High-Fidelity and Diverse Text-to-3D Generation with Variational Score Distillation
> > * [8] Diffusion Schrödinger Bridge Matching
> > * [9] Adversarial Schrödinger Bridge Matching
> > * [10] Flow Straight and Fast: Learning to Generate and Transfer Data with Rectified Flow
> > * [11] Optimal Transport-Guided Conditional Score-Based Diffusion Model
> > * [12] Minimizing Trajectory Curvature of ODE-based Generative Models

---

> > > ### Comment · Reviewer_G3fX · 2024-12-02
> > > **Official Comment by Reviewer G3fX**
> > >
> > > Thank you for your response.
> > >
> > > The main limitations of this work are twofold: first, the novelty of the approach is not sufficiently demonstrated, and second, the reliance on a simple L2 loss appears too heuristic and fails to adequately capture the semantic information necessary for I2I tasks. While the empirical performance suggests that this simple approach yields acceptable results, the justification provided for this performance is insufficient. Please elaborate on the key factors contributing to the reported outcomes.
> > >
> > > Furthermore, the comparison with baseline unpaired I2I translation methods, especially those with accelerated generation, remains inadequate. These issues make the current version not ready for publication, and I will therefore maintain the score as is.

---

### Official Review · Reviewer_v4Tf · 2024-11-04

**Soundness:** 2
**Presentation:** 2
**Contribution:** 2
**Rating:** 5
**Confidence:** 2

**Summary:**

Authors proposed a modified extension of DMD that includes transport cost for unpaired I2I task. They established connection to optimal transport and showed that solution of the soft-constrained RDMD converges to that of the hard-constrained Monge problem. Experimental results on several datasets also prove the effect of the model.

**Strengths:**

Authors prove that solution of the soft-constrained RDMD converges to that of the hard-constrained Monge problem. The proof seems complicated and unfortunately I do not have the mathematical background to fully verify all steps.

**Weaknesses:**

Authors fail to mention related works such as OTCS (Optimal Transport-Guided Conditional Score-Based Diffusion Model). Experiments are also simple and did not outperform baseline in some metrics. Qualitative evaluations are very hard to judge given the 64x64 generated image resolution. Overall it is hard to verify the effectiveness of the proposed method based on the limited experiment results, especially when very related work is missing in the comparison.

**Questions:**

I wonder if there is potential limitations of the method that led to the use of 64x64 image resolution, which is much lower than the standard 256x256 for most of those dataset. Could there be efficiency or computation challenge for scaling up?

---

> ### Author Response · Authors · 2024-11-28
> **Response to Reviewer v4Tf**
>
> We thank the reviewer for the valuable feedback and comments. Below we address weaknesses and the raised questions.
>
> **(1) "Experiments are also simple and didn't outperform baseline in some metrics"**
>
> We would like to note that it is not fully correct to compare against all metrics simultaneously because the unpaired image-to-image translation inherently suggests a trade-off between the quality of generated samples (realism, i.e. measured by FID or other metrics) and the similarity between the generator input and output (i.e. measured by L2/SSIM/PSNR) because the two
> properties interfere with each other.
>
> Therefore, we stress that it is more reasonable to compare the trade-off curves of the methods. We provide the corresponding graphs (FID vs L2/PSNR/SSIM/LPIPS) comparing our method to the baselines in Figure 4, Figure 7 and Figure 10. Here, RDMD achieves a better trade-off than other methods while being one-step (the other methods are not). At the same time, Tables 1 and 2 compare the similarity given the fixed realism (among all runs, we choose ones with the closest FID) and outperform other baselines in at least 2/3 or 3/4 metrics.
>
> **(2) "Qualitative evaluations are very hard to judge given the 64x64 generated image resolution"**
>
> We provide qualitative evaluations for latent space experiments for 256x256 images in Figure 5.
>
> **(3) "Overall it is hard to verify the effectiveness of the proposed method based on the limited experiment results, especially when very related work is missing in the comparison."**
>
> Our experimental results indicate that the method performs better than multi-step diffusion baselines in terms of quality as it gives a better realism-faithfulness tradeoff. We thank the reviewer for pointing to the OTCS [2] paper. In the rebuttal, we add a comparison with a total of 5 baselines, one of which is OTCS. We run it on the AFHQv2 Wild $\rightarrow$ Cat translation problem (along with 3 other baselines: DSBM [3], UOTM [4], DIOTM [5], suggested by the reviewers 4Uuc and q6cb), and obtain the following results:
>
> |          | FID (train) | FID (test) | FID (test $\times 10$) | $\sqrt{L_2}$ | LPIPS     | PSNR      | SSIM      |
> | -------- | ----------- | ---------- | ---------------------- | ------------ | --------- | --------- | --------- |
> | OTCS     | 54.50       | 65.01      | ---                    | **13.71**        | 0.508     | 18.38     | 0.468     |
> | DSBM     | ---         | 25.41      | ---                    | ---          | 0.485     | ---       | ---       |
> | UOTM     | 14.85       | 26.7       | ---                    | 27.39        | 0.509     | 12.14     | 0.250     |
> | DIOTM    | 8.94        | 20.28      | 10.72*                 | 18.69        | 0.465     | 15.66     | 0.496     |
> | **RDMD** | **7.87**    | **18.18**  | **9.24**               | **15.59**    | **0.363** | **19.22** | **0.594** |
>
> While obtaining better $L_2$ and comparable (but still worse) values for other similarity metrics, OTCS fails at producing realistic samples, which is indicated by large FID values and low-quality samples (which we report in Figure 11 of the revised paper version along with the RDMD samples). We also beat all other mentioned baselines in all metrics, thus obtain a method with better faithfulness and realism at the same time.
>
> For comparison with one more baseline, ASBM, please see the general comment or Appendix E of the revised paper (Table 4, Figure 13).
>
> **(4)  "I wonder if there is potential limitations of the method that led to the use of 64x64 image resolution, which is much lower than the standard 256x256 for most of those dataset. Could there be efficiency or computation challenge for scaling up?"**
>
> We stick to 64x64 pixel and 256x256 latent-space experiments due to our constraints in computational resources. The computational complexity of the method is identical to DMD [1]. Therefore, there are no problems with higher dimensions, given the corresponding scale of resources.
>
> **References.**
> - [1] One-Step Diffusion with Distribution Matching Distillation
> - [2] Optimal Transport-Guided Conditional Score-Based Diffusion Model
> - [3] Diffusion Schrödinger Bridge Matching
> - [4] Generative Modeling through the Semi-dual Formulation of Unbalanced Optimal Transport
> - [5] Improving Neural Optimal Transport via Displacement Interpolation

---

### Author Response · Authors · 2024-11-28
**Added comparisons with the suggested baselines**

We sincerely thank the reviewers for their valuable discussion, comments and suggestions. A large part of the remarks is related to the comparison with OT-based methods. Thus, we write a general comment, in which we present comparisons with the suggested baselines.

Among the suggested baselines, we compare with UOTM [1], DIOTM [2], OTCS [3], DSBM [4] and ASBM [5]. We do not directly compare with OTM [6] and NOT [7] since the authors of [1] and [2] demonstrate their inferior performance compared to UOTM and DIOTM. **All the comparisons, including generated samples, can be found in Appendix E of the revised paper.**

We choose two I2I tasks for comparison. First is $64\times64$ AFHQv2 Wild $\rightarrow$ Cat, which is widely studied as a benchmark for these methods (e.g. for DSBM in [8] and DIOTM). Since UOTM is not applied for I2I tasks in the original paper, we modify its official implementation for this scenario and run with the hyperparameters suggested for the larger-scale experiments. We use the official DIOTM implementation from OpenReview and train it with the default Wild $\rightarrow$ Cat hyperparameters to obtain similarity metrics. We also train OTCS on Wild $\rightarrow$ Cat with the hyperparameters used for the main unpaired I2I experiment from the paper (unpaired CelebA deblurring). We do not train DSBM but report its performance measured in [8].

The second I2I task is $64 \times 64$ CelebA Male $\rightarrow$ Female studied, for example, in [2, 5]. Here, we train ASBM with the hyperparameters suggested for the $128 \times 128$ CelebA experiment.

We compare RDMD with OTCS, DSBM, UOTM and DIOTM on **Wild $\rightarrow$ Cat** in the following table, where we beat all the baselines in all the metrics (in both faithfulness and realism), except OTCS, which has a lower $L_2$, but does not fit the target distribution. Figures 11 and 12 from the revised paper version validate the performance qualitatively.

We note that the FID (test $\times 10$) corresponds to the FID calculation protocol used in DIOTM, where the authors generate ten samples per each source (we take this result from the paper and mark it with * ). For a more fair comparison with our one-to-one implementation of RDMD, we adapt this calculation by sampling ten augmentations for each source test sample.

|  Wild $\rightarrow$ Cat        | FID (train) | FID (test) | FID (test $\times 10$) | $\sqrt{L_2}$ | LPIPS     | PSNR      | SSIM      |
| -------- | ----------- | ---------- | ---------------------- | ------------ | --------- | --------- | --------- |
| OTCS     | 54.50       | 65.01      | ---                    | **13.71**        | 0.508     | 18.38     | 0.468     |
| DSBM     | ---         | 25.41      | ---                    | ---          | 0.485     | ---       | ---       |
| UOTM     | 14.85       | 26.7       | ---                    | 27.39        | 0.509     | 12.14     | 0.250     |
| DIOTM    | 8.94        | 20.28      | 10.72*                 | 18.69        | 0.465     | 15.66     | 0.496     |
| **RDMD** | **7.87**    | **18.18**  | **9.24**               | **15.59**    | **0.363** | **19.22** | **0.594** |

Next, we compare RDMD with ASBM on CelebA **Male $\rightarrow$ Female**. Here, we also beat the baseline in all metrics (and obtain higher-quality samples, which can be verified by Figure 13).

|     Male $\rightarrow$ Female     | FID (train) | FID (test) | $\sqrt{L_2}$ | PSNR      | SSIM      |
| -------- | ----------- | ---------- | ------------ | --------- | --------- |
| ASBM     | 22.94       | 31.99      | 15.32        | 17.40     | 0.524     |
| **RDMD** | **12.04**   | **25.6**   | **11.88**    | **20.97** | **0.701** |

**References.**

* [1] Generative Modeling through the Semi-dual Formulation of Unbalanced Optimal Transport
* [2] Improving Neural Optimal Transport via Displacement Interpolation
* [3] Optimal Transport-Guided Conditional Score-Based Diffusion Model
* [4] Diffusion Schrödinger Bridge Matching
* [5] Adversarial Schrödinger Bridge Matching
* [6] Scalable computation of monge maps with general costs
* [7] Neural Optimal Transport
* [8] Schrödinger Bridge Flow for Unpaired Data Translation

---

### Meta-Review · Area_Chair_trte · 2024-12-22

**Metareview:**

This paper extends the diffusion distillation methods, specifically, distribution matching distillation to the unpaired image-to-image translation task. The modification mainly consists of 1) replace the gaussian noise by the source domain distribution, 2) add regularisation related to optimal transport. As pointed out by several reviewers, the novelty of the proposed work is limited. Also, the L2 loss might not be able to capture the semantic difference between domains. Finally, there are several baseline methods that were not included in the comparison. Given these concerns, I would recommend rejecting this paper.

**Additional Comments On Reviewer Discussion:**

The reviewer and authors had a discussion on specific points, but the authors failed to convince the reviewers. Specifically, the remain concerns are not well addressed: 1) the usage of the simple L2 loss might not be appropriate and needs further justification and verifications, 2) the lack of comparison to recent empirical methods on I2I and the datasets need to be more diverse, 3) the novelty of the paper is limited (by reviewer 4UuC, G3fX, and q6cb).

---

### Decision · Program_Chairs · 2025-01-22

Reject